# Robo2 acts in trans to inhibit Slit-Robo1 repulsion in pre-crossing commissural axons

Timothy A Evans[1,2]*[†], Celine Santiago[1][†], Elise Arbeille[1], Greg J Bashaw[1]*

[1]Department of Neuroscience, Perelman School of Medicine, University of Pennsylvania, Philadelphia, United States; [2]Department of Biological Sciences, University of Arkansas, Fayetteville, United States

**Abstract** During nervous system development, commissural axons cross the midline despite the presence of repellant ligands. In *Drosophila*, commissural axons avoid premature responsiveness to the midline repellant Slit by expressing the endosomal sorting receptor Commissureless, which reduces surface expression of the Slit receptor Roundabout1 (Robo1). In this study, we describe a distinct mechanism to inhibit Robo1 repulsion and promote midline crossing, in which Roundabout2 (Robo2) binds to and prevents Robo1 signaling. Unexpectedly, we find that Robo2 is expressed in midline cells during the early stages of commissural axon guidance, and that over-expression of Robo2 can rescue *robo2*-dependent midline crossing defects non-cell autonomously. We show that the extracellular domains required for binding to Robo1 are also required for Robo2's ability to promote midline crossing, in both gain-of-function and rescue assays. These findings indicate that at least two independent mechanisms to overcome Slit-Robo1 repulsion in pre-crossing commissural axons have evolved in *Drosophila*.

*For correspondence: evanst@ uark.edu (TAE); gbashaw@mail. med.upenn.edu (GJB)

[†]These authors contributed equally to this work

Competing interests: The authors declare that no competing interests exist.

## Introduction

The secreted Slit repellents and their Roundabout (Robo) receptors constitute a repulsive axon guidance system whose function is conserved across a wide range of animal taxa including vertebrates, planarians, nematodes, and insects (*Brose and Tessier-Lavigne, 2000*; *Evans and Bashaw, 2012*). Slits are normally expressed at the midline of the central nervous system (CNS), and axons expressing Robo receptors are thus repelled from the midline (*Battye et al., 1999*; *Brose et al., 1999*; *Kidd et al., 1999*). Prior to crossing the midline, commissural neurons in vertebrates and insects prevent premature responsiveness to Slit by regulating the expression and activity of Robo receptors through a variety of mechanisms (*Evans and Bashaw, 2010a*; *Neuhaus-Follini and Bashaw, 2015*). For example, the divergent Robo receptor Robo3/Rig-1 in vertebrates negatively regulates the activity of the Robo1 (Roundabout1) and Robo2 (Roundabout2) receptors in pre-crossing commissural axons in the spinal cord, thereby allowing midline crossing (*Sabatier et al., 2004*). In *Drosophila*, Commissureless (Comm) antagonizes Slit-Robo1 repulsion by preventing the trafficking of the Robo1 receptor to the growth cone, instead diverting newly synthesized Robo1 into the endocytic pathway (*Kidd et al., 1998b*; *Keleman et al., 2002*, *2005*). As commissural axons approach the midline, Comm expression is high, allowing axons to cross the midline (*Keleman et al., 2002*). Once the midline is reached, Comm is down regulated, restoring Robo1-dependent Slit sensitivity and ensuring that commissural axons do not re-cross the midline. Accordingly, loss of Robos or Slits can cause axons to ectopically cross the midline, while loss of Comm or Robo3/Rig1 prevents commissural axons from crossing (*Tear et al., 1996*; *Kidd et al., 1998a*, *1999*; *Long et al., 2004*; *Sabatier et al., 2004*).

**eLife digest** When an animal embryo is developing, nerve pathways grow to connect the left and right halves of the nervous system. These pathways allow coordination between the two sides of the body—which is important for tasks such as walking and swimming. However, in order for these pathways to be properly established, the activity of certain genes must determine whether each nerve fiber (or axon) will stay on one side of the body, or cross the midline to the other side.

It is not fully understood how genes, and the proteins they encode, interact with each other to regulate the crossing of the body's midline, but the process is known to involve proteins called Robo receptors. Robo receptors are a class of proteins found on nerve fibers. Most Robo receptors work to prevent nerve fibers from crossing the midline, but some proteins in this family—including Robo2—can also promote midline crossing. Based on previous studies, it was not clear how Robo2 could have such opposing effects on different sets of nerve cells.

Evans, Santiago et al. have now explored how Robo2 regulates the development of the nervous system of fruit fly embryos, and found that Robo2 promotes midline crossing by inhibiting the activity of a closely related protein called Robo1. Further experiments unexpectedly showed that Robo2 does not promote midline crossing in the cells in which it is produced. Instead, the Robo2 receptor instructs other Robo1-producing nerve cells to cross the midline. These findings reconcile the previous, seemingly paradoxical, observations about the activity of Robo2.

Following on from these findings, one important next step will be to determine exactly how Robo2 can inhibit the activity of Robo1, such that it no longer prevents nerve fibers from crossing the midline. Determining whether similar inhibitory interactions between Robo receptors are important for the development of other tissues in the fruit fly, or in other animals, is another challenge for the future.

---

In *Drosophila*, the three members of the Robo receptor family (Robo1, Robo2, and Robo3) cooperate to control multiple aspects of axon guidance during embryonic development, including midline repulsion of axons and the formation of longitudinal axon pathways at specific mediolateral positions within the nerve cord. Although Robo2 contributes to promoting midline repulsion, gain-of-function genetic experiments suggest that in some contexts Robo2 can also promote midline crossing (*Rajagopalan et al., 2000*; *Simpson et al., 2000b*). More recently, endogenous roles for *robo2* in promoting midline crossing were identified during the guidance of foreleg gustatory neurons in the adult, as well as during the guidance of interneurons in the embryonic CNS (*Mellert et al., 2010*; *Spitzweck et al., 2010*). Robo2's pro-crossing role in the embryo is highlighted in *frazzled* and *Netrin* mutant backgrounds, in which midline attraction is partially compromised (*Spitzweck et al., 2010*). In the absence of Netrin-dependent midline axon attraction, loss of *robo2* (but not *robo1* or *robo3*) leads to a dramatic disruption in midline crossing that is far more severe than the complete loss of Netrins, indicating that *robo2* likely acts in parallel to Netrin-Fra to promote midline crossing (*Spitzweck et al., 2010*).

In a complementary series of gain-of-function experiments using a panel of chimeric receptors comprising different regions of Robo1 and Robo2 fused together, we have previously shown that Robo2's ability to promote ectopic midline crossing correlates with the presence of the first and second immunoglobulin-like domains (Ig1 and Ig2) within its extracellular domain (*Evans and Bashaw, 2010b*). Consistent with these observations, replacing endogenous Robo2 by homologous recombination with chimeric receptors, in which the cytoplasmic domains of the Robo1 and Robo2 receptors were exchanged, reveals that the Robo2-1 chimeric receptor (containing the extracellular region of the Robo2 receptor) can rescue the commissural guidance defect observed in *Netrin*, *robo2* mutants more effectively than the reciprocal chimeric receptor (*Spitzweck et al., 2010*). However, the mechanism by which Robo2 promotes midline crossing remains unclear. We can envision two alternative models that could account for Robo2's role in promoting midline crossing of commissural axons. First, Robo2 may act as an attractive receptor to signal midline attraction in response to a ligand produced by midline glia, analogous to Frazzled/Deleted in Colorectal Cancer (DCC)'s role in Netrin-dependent midline attraction. Indeed, a role for Robo2 in mediating attractive responses to Slit has been described in the context of muscle cell migration (*Kramer et al., 2001*). Alternatively,

Robo2 may antagonize Slit-Robo1 repulsion by preventing Robo1 from signaling in response to midline-derived Slit, similar to the proposed role of Robo3/Rig-1 in pre-crossing commissural axons in the vertebrate spinal cord (*Figure 1*). Although Comm is an essential regulator of Robo1 activity in *Drosophila*, low levels of Robo1 escape Comm-dependent sorting and can be detected on commissural axons, raising the question of whether and how the activity of these Robo1 receptors is regulated (*Kidd et al., 1998a*).

Here, we show that in addition to its cell-autonomous role in midline repulsion, Robo2 acts non-autonomously to promote midline crossing by inhibiting canonical Slit-Robo1 repulsion and offer insights into the molecular and cellular mechanisms underlying this activity of Robo2. We find that the cytoplasmic domain of Robo2 is dispensable for its pro-crossing role, suggesting that Robo2 does not transduce a midline attractive signal, and that Robo2 over-expression can suppress *comm* mutants, supporting a model in which Robo2 antagonizes Slit-Robo1 repulsion. Moreover, Robo2 can bind to Robo1 in *Drosophila* embryonic neurons, and this biochemical interaction, like Robo2's pro-crossing role, correlates with the presence of Ig1 and Ig2. Surprisingly, we observe that Robo2 is able to promote midline crossing of axons non-cell autonomously when mis-expressed in midline cells, and we further show that Robo2 is expressed in midline glia and neurons during the early stages of commissure formation. Finally, we find that restoring Robo2 expression in midline cells can rescue midline crossing of axons in *robo2, fra* double mutants and that this rescue activity is dependent on Ig1 and Ig2. Together, our results indicate that Robo2 acts non-autonomously to bind to Robo1 and prevent Slit-Robo1 repulsion in pre-crossing commissural axons. This model accounts for Robo2's seemingly paradoxical roles in both promoting and inhibiting midline crossing and explains how the small amount of Robo1 present on pre-crossing commissural axons might be prevented from responding to Slit.

## Results

The midline attractive ligand Netrin and its receptor Frazzled (Fra) are the only known attractive ligand-receptor pair in *Drosophila*, yet many commissural axons still cross the midline in the absence of attractive Netrin-Frazzled signaling (*Kolodziej et al., 1996*; *Mitchell et al., 1996*). It has recently been demonstrated that the Robo family receptor Robo2 acts independently of Netrin and Fra to promote midline crossing, through an as yet unknown mechanism (*Spitzweck et al., 2010*). In *robo2, fra* double mutants, midline crossing of commissural axons is severely compromised, leading to thin or absent commissures, a phenotype that is qualitatively and quantitatively more severe than loss of *fra* alone (*Figure 1*). This phenotype can be observed by staining the entire axon scaffold with anti-HRP antibodies (*Figure 1A–D*) or by labeling a subset of commissural axons using *eg-GAL4* (*Garbe et al., 2007*; *O'Donnell and Bashaw, 2013*) in *robo2, fra* double mutants (*Figure 1F–I*). To quantify the midline crossing defects, we scored the number of segments in which the EW axons, which normally cross the midline in the posterior commissure, fail to cross (*Figure 1*, histogram). We find that in *robo2, fra* double mutants approximately 70% of EW axons fail to cross the midline, compared to around 30% in *fra* mutants. Analysis of cell fate markers including Eg, even-skipped and zfh1 revealed no gross differences in segmentation and neuronal differentiation in *robo2, fra* double mutants, and although the cell bodies of the EW neurons were sometimes displaced, they were easily identifiable (data not shown). Importantly, restoring Robo2 expression by introducing one copy of an 83.9 kb *robo2* bacterial artificial chromosome (BAC) transgene that includes the entire 40 kb *robo2* transcription unit in this background significantly rescues the EW axon crossing defects (*Figure 1E,J*), confirming that this is a *robo2*-dependent phenotype.

## Robo2's pro-crossing activity does not require its cytoplasmic domain

If Robo2 were to act as a midline attractive receptor (*Figure 1*, model 1), its cytoplasmic domain would likely be required for midline attraction. To test whether the Robo2 cytoplasmic domain contributes to its pro-crossing activity, we tested whether a truncated Robo2 receptor lacking its cytoplasmic domain (Robo2ΔC) could promote midline crossing when mis-expressed in embryonic neurons. We found that, as with full-length Robo2, pan-neural mis-expression of Robo2ΔC (with *elav-GAL4*) produced strong ectopic crossing of FasII-positive axons in the embryonic CNS (*Figure 2*). Indeed, the Robo2ΔC mis-expression phenotype was stronger than full-length Robo2. In contrast, pan-neural over-expression of Robo1 did not generate ectopic crossing (*Figure 2*). In these experiments, all UAS-Robo transgenes are expressed from the same genomic insertion site in order to

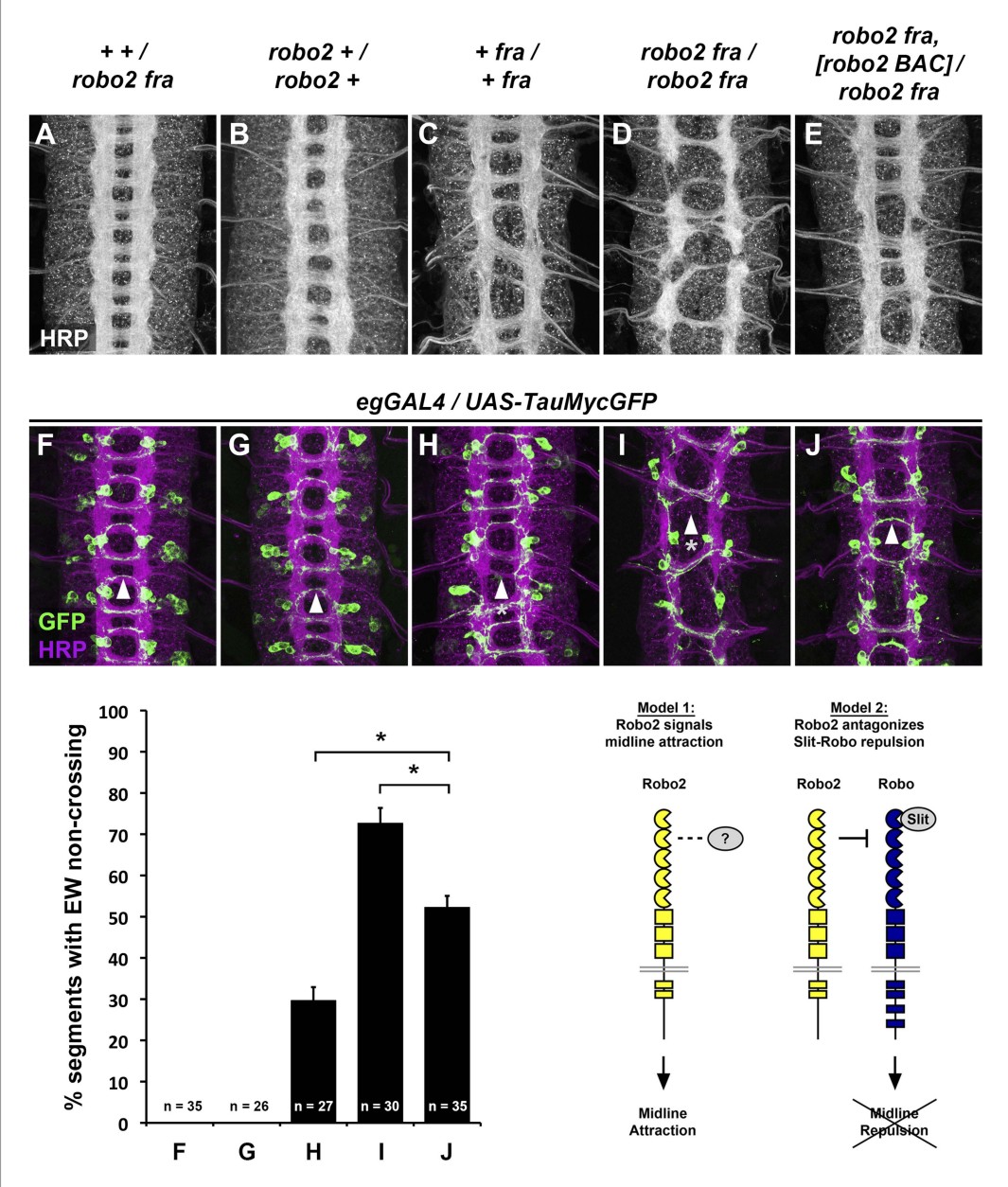

**Figure 1**. Robo2 commissural guidance defects are rescued by a Robo2 BAC transgene. (**A–E**) Stage 17 Drosophila embryos of the indicated genotypes stained with anti-HRP antibodies to label all CNS axons. (**F–J**) Stage 15–16 embryos of the indicated genotypes carrying *eg-GAL4* and *UAS-TauMycGFP* transgenes, stained with anti-HRP and anti-GFP antibodies. Anti-GFP labels cell bodies and axons of the eagle neurons (EG and EW) in these embryos. (**A** and **F**) Embryos heterozygous for both *frazzled* (*fra*) and *robo2* display a wild-type arrangement of longitudinal and commissural axon pathways, and axons of the EW neurons cross the midline in the posterior commissure in 100% of segments (arrowhead). (**B** and **G**) *robo2* mutants (*robo2^123^/robo2^33^*) display a mildly disorganized axon scaffold, but no detectable defects in EW crossing. (**C** and **H**): *fra* mutants (*fra^3^/fra^4^*) display thinning commissures indicative of decreased midline crossing, and the EW axons fail to cross the midline in 30% of abdominal segments (arrowhead with asterisk). (**D** and **I**) Simultaneous removal of *robo2* and *fra* (*robo2^123^, fra^3^/robo2^135^, fra^4^*) strongly enhances the midline crossing defects seen in *fra* single mutants. (**E** and **J**) Midline crossing is partially restored in *robo2, fra* double mutants carrying one copy of an 83.9-kb *robo2* BAC transgene. The overall organization of the axon scaffold approaches that seen in *fra* single mutants, and EW axon crossing defects are significantly rescued, although not completely restored to the level seen in *fra* single mutants. Histogram quantifies EW midline crossing defects in the genotypes shown in (**F–J**). Error bars represent s.e.m. *n*, number of embryos scored for each genotype (*$p < 0.0001$). Bottom right: Two possible models for how Robo2 might promote midline crossing of commissural axons. Left, Robo2 may act as a midline attractive receptor to promote midline crossing in response to an

*Figure 1. continued on next page*

*Figure 1. Continued*

unidentified ligand, analogous to Fra's role as an attractive Netrin receptor. Right, Robo2 may antagonize canonical Slit-Robo1 repulsive signaling to down-regulate midline repulsion and thus allow Robo1-expressing axons to cross the midline.

ensure that they are expressed at similar levels. Importantly, the pro-crossing activity of Robo2ΔC is unlikely to be caused solely by a dominant-negative effect of the truncated receptor, as a similarly truncated form of Robo1 (Robo1ΔC) has a qualitatively weaker ectopic crossing phenotype when combined with *elav-GAL4* (*Figure 2*). Robo2ΔC expression, unlike Robo1ΔC, leads to ectopic crossing of all of the ipsilateral FasII axon bundles and also results in many segments exhibiting a *slit*-like phenotype (*Figure 2*). Due to the strong phenotypic effects of the targeted insertion lines of Robo1ΔC and Robo2ΔC, we also compared the phenotypes generated by lower levels of expression of the two truncated receptors using standard UAS inserts and observed that Robo2ΔC is significantly more potent at driving ectopic midline crossing than comparable levels of the Robo1ΔC receptor (*Figure 3*). Together these observations indicate that the pro-crossing activity of Robo2 is independent of the cytoplasmic domain and argue against the idea that Robo2 promotes midline crossing by signaling attraction.

## Robo2's pro-crossing activity does not strictly depend on Slit binding

Slit is the canonical ligand for Robo family receptors, and all three *Drosophila* Robos can bind to the single *Drosophila* Slit (*Howitt et al., 2004*). To test whether Robo2's pro-crossing activity depends on its ability to bind Slit, we deleted the canonical Slit-binding domain (the first immunoglobulin-like domain: Ig1) from Robo2. As predicted by previous in vitro binding studies using *Drosophila* Robo1 (*Brose et al., 1999*; *Fukuhara et al., 2008*), we found that deleting Ig1 from Robo2 prevented Slit binding in cultured *Drosophila* cells (*Figure 4* and *Figure 4—figure supplement 1*). Pan-neural over-expression of Robo2 produces a phenotype in which some axons are repelled from the midline, and some axons ectopically cross the midline, reflecting Robo2's two opposing activities in regulating midline crossing. As expected, deleting the Ig1 domain prevents Robo2 from signaling midline repulsion in vivo, both broadly in all neurons (when expressed pan-neurally with *elav-GAL4*) (*Figure 5*) and in a subset of commissural neurons (the EW neurons, labeled by *eg-GAL4*) (*Figure 4*), confirming that Slit binding is required for Robo2-mediated repulsion. In contrast, we found that the Robo2 receptor lacking Ig1 retained a partial ability to promote ectopic midline crossing of FasII-positive axons, indicating that the pro-crossing activity of Robo2 does not strictly depend on its ability to bind Slit (*Figure 5*). Notably, the ectopic crossing phenotype produced by Robo2ΔIg1 mis-expression was significantly weaker than that caused by mis-expression of full-length Robo2 (*Figure 5*). This result suggests that the Ig1 domain contributes to, but is not strictly required for, promotion of midline crossing by Robo2.

## Robo2's Ig2 domain is required for its pro-crossing activity

We have previously shown that Robo2's pro-crossing activity is conferred at least in part by its Ig2 domain: replacing the Ig1–Ig2 region of Robo1 with the equivalent region from Robo2 (Robo1$^{R2Ig1+2}$) confers Robo2-like pro-crossing activity to Robo1 (*Evans and Bashaw, 2010b*). Further, replacing Ig1–Ig2 of Robo2 with Robo1 Ig1–Ig2 (Robo2$^{R1Ig1+2}$) abolishes its pro-crossing activity (*Evans and Bashaw, 2010b*). To directly test whether Ig2 is necessary for Robo2 to promote midline crossing, we generated a Robo2 receptor lacking Ig2 but with all other Ig domains intact (Robo2ΔIg2). We found that deleting Robo2's Ig2 domain did not interfere with Slit binding (*Figure 4*), nor did it affect Robo2's ability to signal repulsion in commissural neurons (*Figure 4*). However, deletion of Robo2's Ig2 domain strongly disrupted its ability to promote ectopic midline crossing (*Figure 5*). The low level of ectopic crossing induced by Robo2ΔIg2 was indistinguishable from that caused by Robo3, a related receptor that does not share Robo2's pro-midline crossing activity (*Figure 5*). These results contrast with those observed with Robo2ΔIg1, which lacks Slit-dependent midline repulsive activity but retains some pro-midline crossing activity (*Figures 4, 5*). These data indicate that Robo2's Ig2 domain is essential for promoting midline crossing when Robo2 is mis-expressed in all neurons. In these

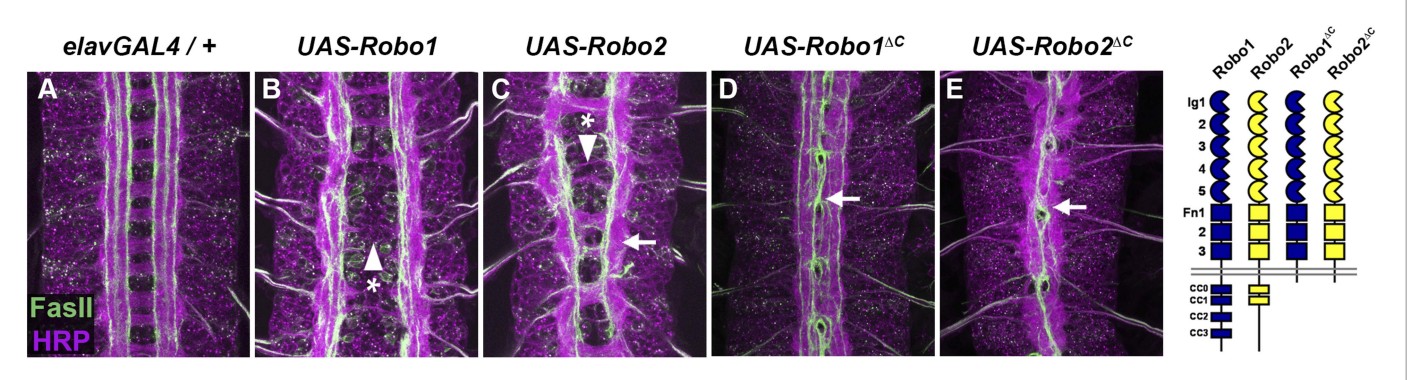

**Figure 2**. Robo2 can promote midline crossing independent of its cytoplasmic domain. (**A–E**) Stage 17 embryos carrying *elav-GAL4* and the indicated *UAS-Robo* transgenes, stained with anti-HRP (magenta) and the longitudinal pathway marker anti-FasciclinII (FasII; green). (**A**) Embryos carrying *elav-GAL4* alone exhibit a wild-type arrangement of axon pathways, including distinct anterior and posterior commissures and three FasII-positive longitudinal pathways that do not cross the midline. (**B**) In *elav-GAL4/UAS-Robo1* embryos, commissure formation is strongly impaired, and no ectopic midline crossing of FasII-positive axons is observed. (**C**) Mis-expression of Robo2 with *elav-GAL4* produces a biphasic phenotype, where some segments appear nearly commissureless (arrowhead with asterisk) while others exhibit ectopic crossing reminiscent of *robo1* mutants (arrow). See *Figure 5* for quantification of ectopic crossing in *elav-GAL4/UAS-Robo2* embryos. (**D** and **E**) Mis-expression of truncated forms of Robo1 (Robo1ΔC) or Robo2 (Robo2ΔC) with *elav-GAL4* induces ectopic crossing in 100% of segments, although the Robo2ΔC mis-expression phenotype is qualitatively more severe than Robo1ΔC. In *elav-GAL4/UAS-Robo1ΔC* embryos (**D**) only the medial FasII pathway crosses the midline and the axon scaffold overall exhibits a *robo1*-like appearance, while in *elav-GAL4/UAS-Robo2ΔC* embryos (**E**) all three FasII-positive pathways collapse at the midline in nearly every segment and the axon scaffold appears *slit*-like. All UAS-Robo transgenes shown here were inserted into the same genomic location (86FB) to ensure equivalent expression levels.

experiments, all UAS-Robo transgenes are expressed from the same genomic insertion site in order to ensure that they are expressed at similar levels. In addition, we assayed the protein localization and expression levels of Robo2 and its deletion variants and observed comparable surface expression in cultured S2R+ cells in vitro, as well as comparable expression levels and localization in CNS axons in vivo (*Figure 4—figure supplement 2*).

## Robo2 can promote crossing non-autonomously

Midline crossing is strongly reduced in *robo2, fra* double mutants, and pan-neural mis-expression of Robo2 can promote ectopic midline crossing. However, it is unclear whether Robo2 acts autonomously or non-autonomously to promote midline crossing. The ectodomain-dependent nature of Robo2's pro-crossing activity and our pan-neural mis-expression assays do not distinguish between these possibilities. Indeed, *elav-GAL4* is transiently expressed in midline glia as well as post-mitotic neurons, preventing us from ruling out a non-neuronal contribution to the observed mis-expression phenotypes. Notably, we have never observed a clearly cell-autonomous pro-crossing phenotype caused by Robo2. We have previously described a series of experiments in which we mis-expressed full-length, truncated, or chimeric receptor variants of Robo1 and Robo2 in a subset of ipsilateral neurons (the apterous neurons, labeled by *ap-GAL4*) (*Evans and Bashaw, 2010b*). In contrast to the very different phenotypes caused by pan-neural mis-expression of these two truncated receptors (where Robo2ΔC is much more potent at inducing midline crossing than Robo1ΔC), Robo1ΔC and Robo2ΔC induce similar low levels of ectopic crossing when expressed in the apterous neurons (*Figure 6A*). We interpret this as a cell-autonomous dominant-negative effect of these truncated receptors.

Full-length Robo2 is unable to autonomously promote midline crossing of the apterous axons (*Evans and Bashaw, 2010b*). Instead, Robo2 mis-expression redirects apterous axons to lateral regions of the neuropile. In the course of examining this lateral positioning activity of Robo2, we mis-expressed Robo2 in a second class of longitudinal interneurons: those labeled by *Hb9-GAL4*. Intriguingly, we observed two distinct phenotypes in embryos where *Hb9-GAL4* drives Robo2 expression. First, *Hb9*-positive axons were shifted to more lateral positions within the neuropile. Second, *Hb9*-negative FasII-positive axons ectopically crossed the midline (*Figure 6B*). These results

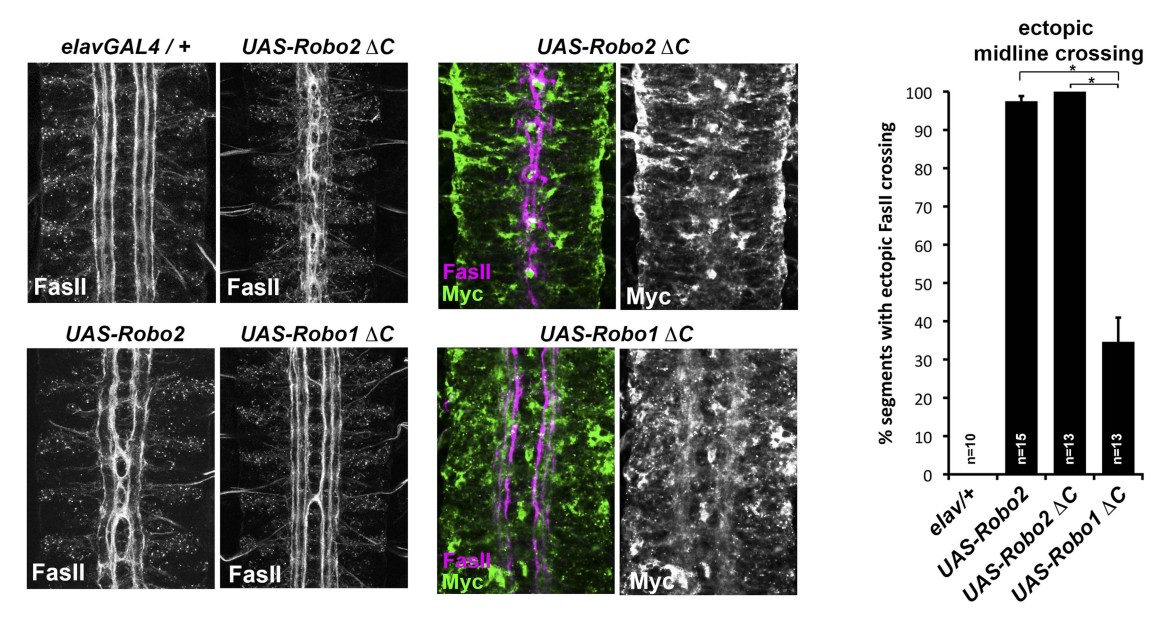

**Figure 3**. Comparison of Robo1ΔC and Robo2ΔC gain of function activities. Since the effects of expressing ΔC transgenes in the 86Fb insertion site are too potent to allow quantitative comparison, we used traditional UAS insertion lines that are expressed at lower and comparable levels (right panels, anti-Myc is shown in green and anti-FasII in magenta) to compare activities of Robo1ΔC and Robo2ΔC. In embryos expressing only an *elav-GAL4* transgene (top left), FasII axons appear wild-type and remain ipsilateral. Mis-expression of Robo2 leads to a high level of ectopic crossing. Robo2ΔC expression results in a much greater degree of ectopic midline crossing than does Robo1ΔC. Segments with ectopic midline crossing of FasII axons are quantified on the right. Significance was assessed by multiple comparisons using the Student's *t*-test and a Bonferonni correction (*p < 0.001). Error bars represent s.e. m. *n*, number of embryos scored for each genotype.

suggest that Robo2 can autonomously specify the lateral position of *Hb9*-positive axons while non-autonomously instructing FasII axons to cross the midline. We note that *Hb9-GAL4* expression initiates earlier than *ap-GAL4* and includes a larger number of neurons, including some located near the CNS midline (such as the RP motor neurons), suggesting the possibility that early midline-proximal expression of Robo2 accounts for the non-autonomous effect observed with *Hb9-GAL4*.

To more explicitly test whether Robo2 can promote midline crossing non-autonomously, we used *slit-GAL4* to drive Robo2 expression in midline glia and neurons. We found that mis-expression of Robo2 or Robo2ΔC in midline cells caused many FasII-positive axons which do not express *slit-GAL4* to ectopically cross the midline, confirming that Robo2 can act non-autonomously to promote midline crossing of axons, and that this effect does not depend on the cytoplasmic domain (*Figure 7*). We observed a significantly milder effect with mis-expression of Robo1, suggesting that Robo2's non-cell autonomous activity is not solely a consequence of Slit titration (*Figure 7*). Moreover, this non-cell autonomous activity of Robo2 appears to be Ig1/Ig2-dependent, as Robo1[R2Ig1+2] but not Robo2[R1Ig1+2] promoted strong ectopic midline crossing when expressed using *slit-GAL4* (*Figure 7*). In addition, expression of Robo2 variants missing either Ig1 or Ig2 with *slit-Gal4* did not result in any ectopic midline crossing (*Figure 7*). The requirement for both Ig1 and Ig2 in this context contrasts with our findings with pan-neural mis-expression, in which Robo2ΔIg1 retained some pro-crossing activity. However, it is worth noting that the phenotype generated by *elav-GAL4* mis-expression of Robo2 is stronger than that generated by *slit-GAL4*, perhaps because *slit-GAL4* is expressed in a much smaller number of cells.

### *robo2* is expressed in midline glia and neurons during commissure formation

Robo2 can promote midline crossing when expressed in a subset of embryonic neurons and glia, and endogenous *robo2* contributes to midline crossing of commissural axons. During embryogenesis,

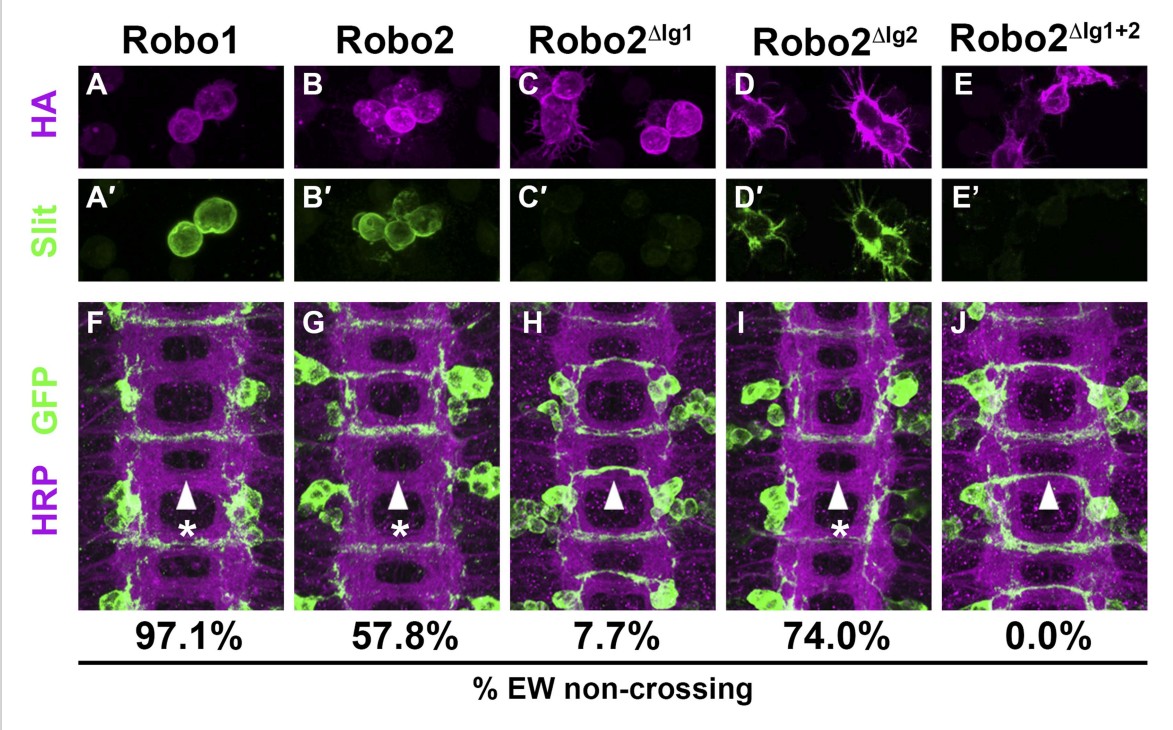

**Figure 4.** Slit binding and Robo gain of function. (**A–E**) Slit-conditioned media was collected and used to treat cells expressing the indicated HA-tagged receptors. Receptor expression is shown with anti-HA in the top panels (magenta) and anti-Slit staining is shown in the bottom panels (green, **A′–E′**). Robo1 (**A**), Robo2 (**B**), and Robo2ΔIg2 (**D**) bind efficiently to Slit, while little to no binding is detected in cells expressing Robo2ΔIg1 (**C**) or Robo2ΔIg1+2 (**E**). (**F–J**) Stage 16 embryos expressing the indicated transgene in the Eg commissural interneurons. HRP labels the axon scaffold (magenta) and anti-GFP labels the Eg neurons. The percentages under each panel indicate the percentage of EW axons that fail to cross the midline in each condition. Expression of Robo1 (**F**), Robo2 (**G**), and Robo2ΔIg2 (**I**) all lead to strong disruption of midline crossing, while expression of Robo2ΔIg1 (**H**), and Robo2ΔIg1+2 (**J**) result in little to no crossing defects.

The following figure supplements are available for figure 4:

**Figure supplement 1.** Quantification of relative fluorescence intensity of HA and Slit antibody staining in S2R+ cells transfected with HA-tagged Robo2 proteins, and treated with Slit-conditioned media.

**Figure supplement 2.** Robo2 transgenes are localized to axons and expressed at equivalent levels in vivo and are present at the surface of S2R+ cells in vitro.

*robo2* expression is dynamically regulated: it is broadly expressed in neurons during early stages of CNS development, including transient expression in a number of ipsilateral pioneer neurons, and later becomes restricted to neurons whose axons form longitudinal pathways in the lateral regions of the neuropile (*Simpson et al., 2000a*). To gain additional insight into Robo2's role in promoting midline crossing of commissural neurons, we examined *robo2* mRNA and protein expression in embryos during the early stages of axon path finding, when the first commissural axons are crossing the midline (stages 12–13). Using fluorescent mRNA in situ hybridization, we were able to detect *robo2* mRNA expression in cells labeled by *slit-GAL4* in late stage 12 embryos, around the time that pioneer commissural axons are crossing the midline (*Figure 8B*). *robo2* mRNA expression persists through the end of stage 13, but is no longer detectable by stage 14; thus, midline expression of *robo2* coincides with the time when most commissural axons are crossing the midline (*Figure 8*, *Figure 8—figure supplement 1*). Moreover, a *robo2-GAL4* enhancer-trap insertion is expressed in midline glia at this time, as detected by anti-GFP staining in *robo2-GAL4, UAS-TauMycGFP* embryos (*Figure 8A*). Expression of *UAS-HARobo2* with *robo2-Gal4* and detection of transgenic Robo2 with anti-HA reveals an expression pattern that closely resembles the endogenous pattern of Robo2 protein, suggesting

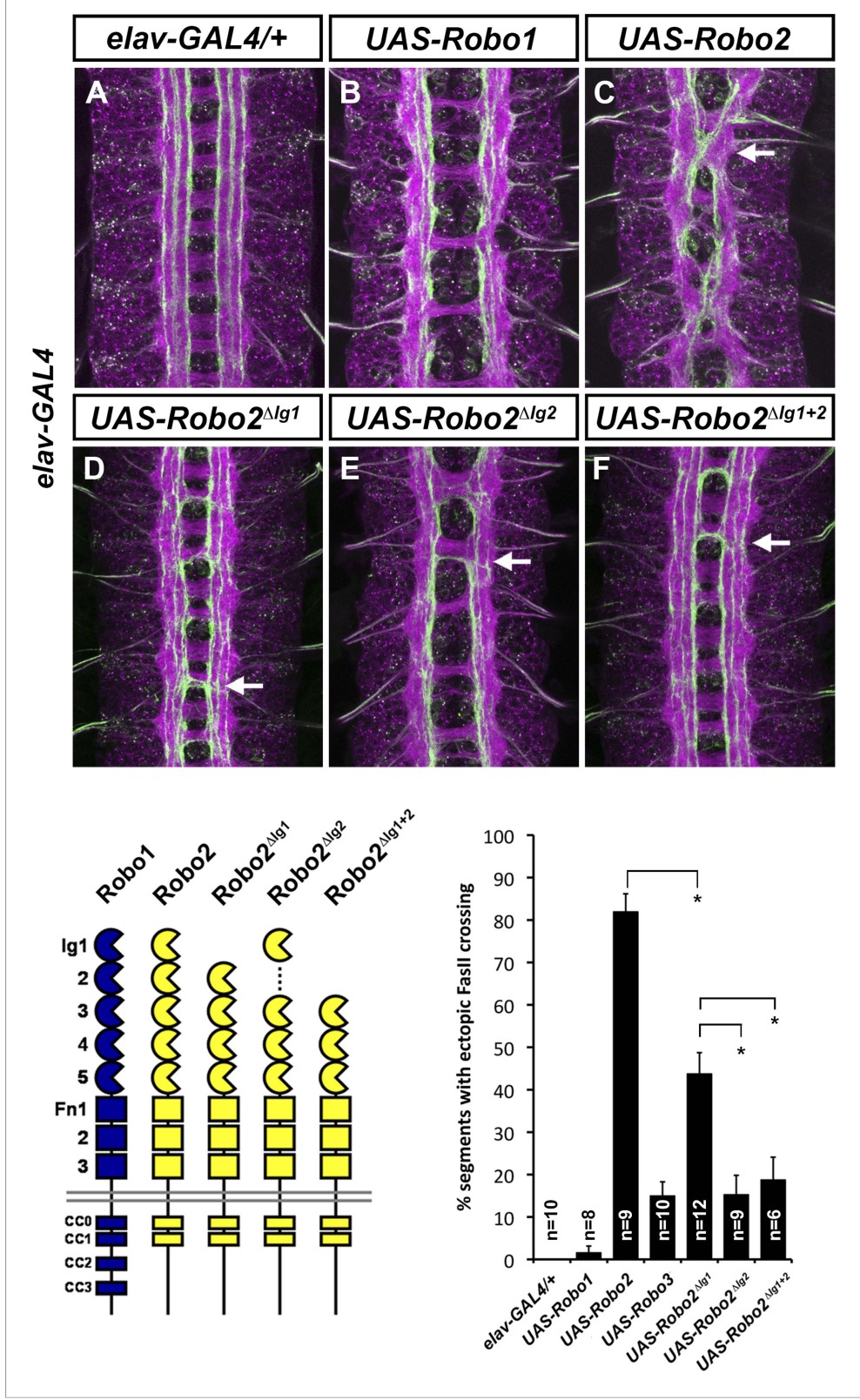

**Figure 5**. Robo2's pro-crossing activity depends on its Ig2 domain. (**A–F**) Stage 17 embryos carrying *elav-GAL4* and the indicated *UAS-Robo* transgenes, stained with anti-HRP and anti-FasII. (**A**) Embryos carrying *elav-GAL4* alone exhibit a wild-type arrangement of axon pathways, including three FasII-positive longitudinal pathways that do not

*Figure 5. continued on next page*

*Figure 5. Continued*

cross the midline. (**B**) Robo1 does not promote midline crossing of FasII-positive axons when misexpressed in all neurons with *elav-GAL4*. (**C**) Misexpression of full-length Robo2 induces ectopic midline crossing in over 80% of segments (arrow). (**D**) Deleting the Ig1 domain (Robo2$^{\Delta Ig1}$) disrupts Slit binding but does not completely prevent Robo2 from promoting midline crossing. (**E** and **F**) Robo2 receptors lacking the Ig2 domain (Robo2$^{\Delta Ig2}$) or both the Ig1 and Ig2 domains (Robo2$^{\Delta Ig1+2}$) are unable to promote ectopic midline crossing above background levels (both are comparable to Robo3; see histogram). Schematics show domain composition of receptors shown in (**A–F**). All UAS-Robo transgenes shown here were inserted into the same genomic location (86FB) to ensure equivalent expression levels. Histogram quantifies ectopic midline crossing in the indicated genotypes. Significance was assessed by multiple comparisons using the Student's *t*-test and a Bonferonni correction (*p < 0.01). Error bars represent s.e.m. *n*, number of embryos scored for each genotype.

that the *robo2-GAL4* faithfully reports Robo2 expression (data not shown). In addition, we could detect weak expression of Robo2 protein produced by an HA-tagged knock-in allele of *robo2* (*Spitzweck et al., 2010*) in a subset of *slit-GAL4* expressing cells at stage 12, confirming that Robo2 protein is produced in midline cells during the stages of commissural axon path finding, and raising the possibility that Robo2 endogenously acts in these cells to promote midline crossing of commissural axons (*Figure 8B*).

## Midline expression of Robo2 rescues the commissural defects in *fra*, *robo2* mutants

Robo2 can promote midline crossing non-autonomously, and endogenous *robo2* expression can be detected in *slit-GAL4*-expressing cells as well as in contralateral and ipsilateral neurons during the initial stages of commissure formation (*Figure 8* and data not shown). Our ability to partially rescue midline crossing in *robo2*, *fra* double mutants with the *robo2* BAC confirms that this is a *robo2*-specific

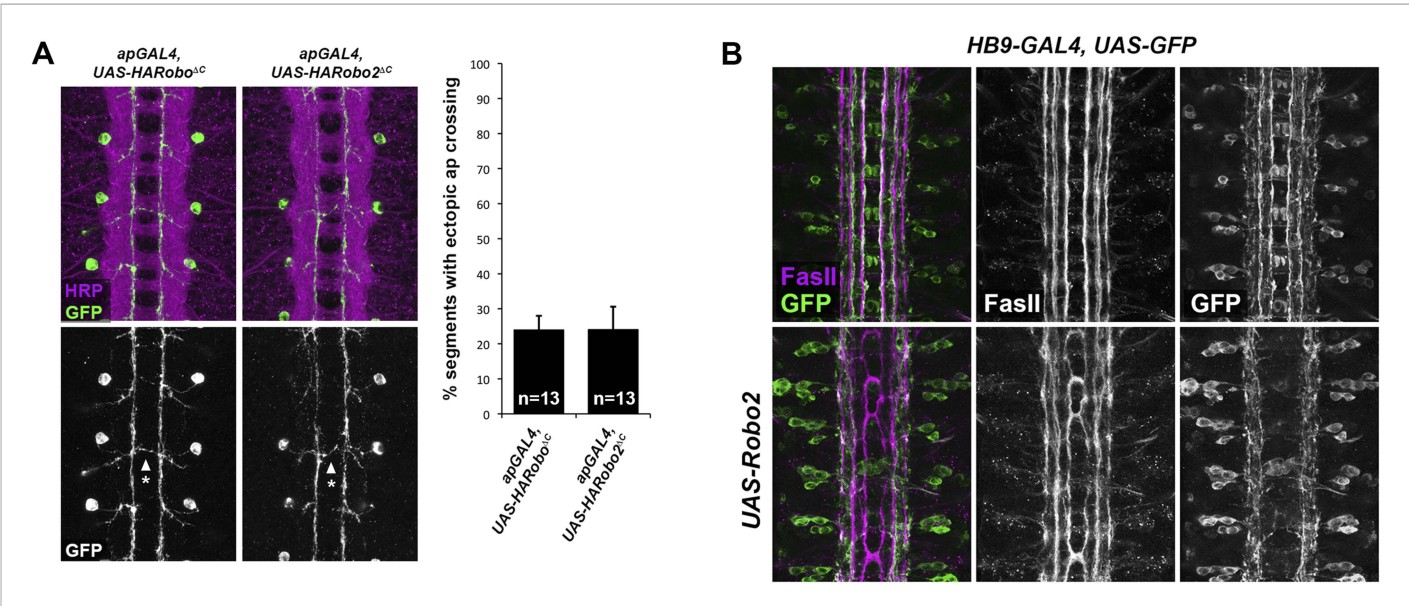

**Figure 6**. Robo2 acts cell non-autonomously to promote midline crossing in ipsilateral neurons. (**A**) Stage 17 embryos stained with anti-HRP (magenta) and anti-GFP (green) antibodies. Anti-GFP labels the apterous (ap) cell bodies and axons, which normally project ipsilaterally. Mis-expression of Robo2ΔC in ap neurons results in a mild ectopic crossing phenotype, which is similar to the effect of Robo1ΔC (arrowheads with asterisks). Segments with ectopic crossing of ap axons are quantified in the histogram. (**B**) Stage 17 embryos stained with anti-FasII (magenta) and anti-GFP (green) antibodies. Anti-GFP labels the axons of *hb9-GAL4* expressing cells. Mis-expression of Robo2 with *hb9-GAL4* results in a lateral shift of *hb9-Gal4+* axons, and causes FasII+ axons that do not express *hb9-GAL4* to ectopically cross the midline.

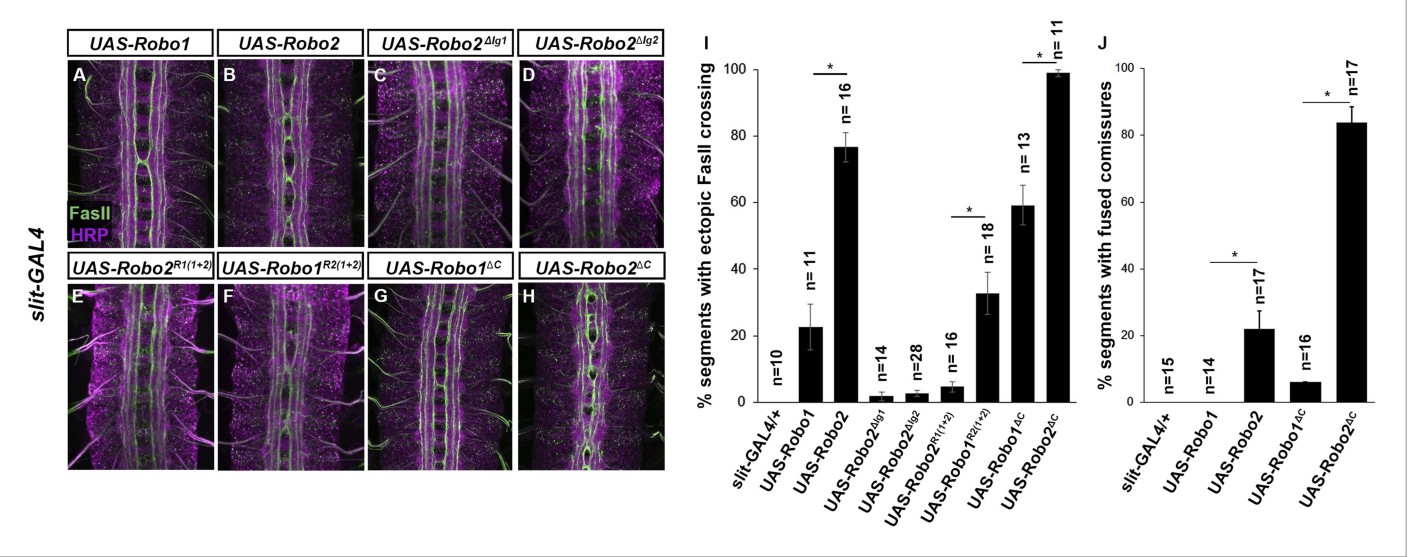

**Figure 7**. Robo2 can promote crossing non cell-autonomously. (**A–D**) Stage 17 embryos stained with anti-HRP (magenta) and anti-FasII (green). (**A** and **B**) Mis-expression of Robo1 (**A**) in midline cells using *slit-GAL4* results in a mild ectopic crossing phenotype. In contrast, mis-expression of Robo2 (**B**) produces a much stronger effect, as indicated by quantification of ectopic FasII crossing in the histogram (**I**). (**C** and **D**) Mis-expression of either Robo2ΔIg1 (**C**) or Robo2ΔIg2 (**D**) with *slit-GAL4* does not produce ectopic crossing of FasII axons. (**E** and **F**) Consistent with requirement of Robo2's first two IG domains, the chimeric protein Robo1$^{R2IG(1+2)}$ produces an ectopic crossing phenotype (**F**), whereas Robo2$^{R1(IG1+2)}$ has no effect (**E**). (**G** and **H**) Mis-expression of Robo2ΔC with *slit-GAL4* also results in severe ectopic crossing defects (**H**) that are much stronger than those observed with Robo1ΔC (**G**), as indicated by quantification of ectopic FasII crossing (**I**) and fused commissures observed in anti-HRP stained embryos (**J**). All UAS-Robo transgenes were inserted into the same genomic location (86FB). Significance was assessed by multiple comparisons using the Student's *t*-test and a Bonferonni correction (*p < 0.001). Error bars represent s.e.m. *n*, number of embryos scored for each genotype.

phenotype, but does not address in which cells *robo2* acts to instruct commissural axons to cross the midline. To address this question, we attempted to rescue *robo2*'s endogenous pro-crossing activity by restoring *robo2* expression in restricted subsets of cells in *robo2, fra* double mutants. We first expressed Robo2 in the commissural EW neurons (using *eg-GAL4*) in *robo2, fra* double mutants. We found that neither full-length Robo2 nor Robo2ΔC can rescue midline crossing when expressed autonomously in the EW neurons (*Figure 9*), suggesting that Robo2 may not act cell autonomously to promote midline crossing. We next attempted to rescue midline crossing in *robo2, fra* double mutants by expressing Robo2 using *slit-GAL4*. Strikingly, we found that driving Robo2 expression in these cells significantly restores posterior commissure formation, as measured by anti-HRP staining (*Figure 8C–G*). Furthermore, this effect is dependent on Ig1 and Ig2 (*Figure 8C–G*). Cell-type specific loss of function experiments will be necessary to confirm the site of Robo2's endogenous activity, and our attempts to recapitulate the *robo2, fra* phenotype by over-expression of RNAi transgenes have so far been unsuccessful, likely because of the well known difficulty of achieving sufficient knockdown in embryonic stages. Nevertheless, our results suggest that Robo2 promotes midline crossing non-cell autonomously, and may act in midline glia and neurons, where it is expressed during the stages of commissural axon path finding, to promote midline crossing of commissural axons.

## Robo2 can antagonize Slit-Robo1 repulsion

Robo2 can promote midline crossing of axons independently of its cytoplasmic domain and its Slit-binding Ig1 domain, suggesting that Robo2 does not promote crossing by acting as an attractive signaling receptor or solely by titrating Slit. Does Robo2 antagonize Slit-Robo1 repulsion through another mechanism? In order to test this hypothesis, we took advantage of *comm* mutants, which provide a genetic background in which hyperactive Slit-Robo1 signaling completely prevents midline crossing. In *comm* mutants, endogenous Robo1 is inappropriately trafficked to the growth cone

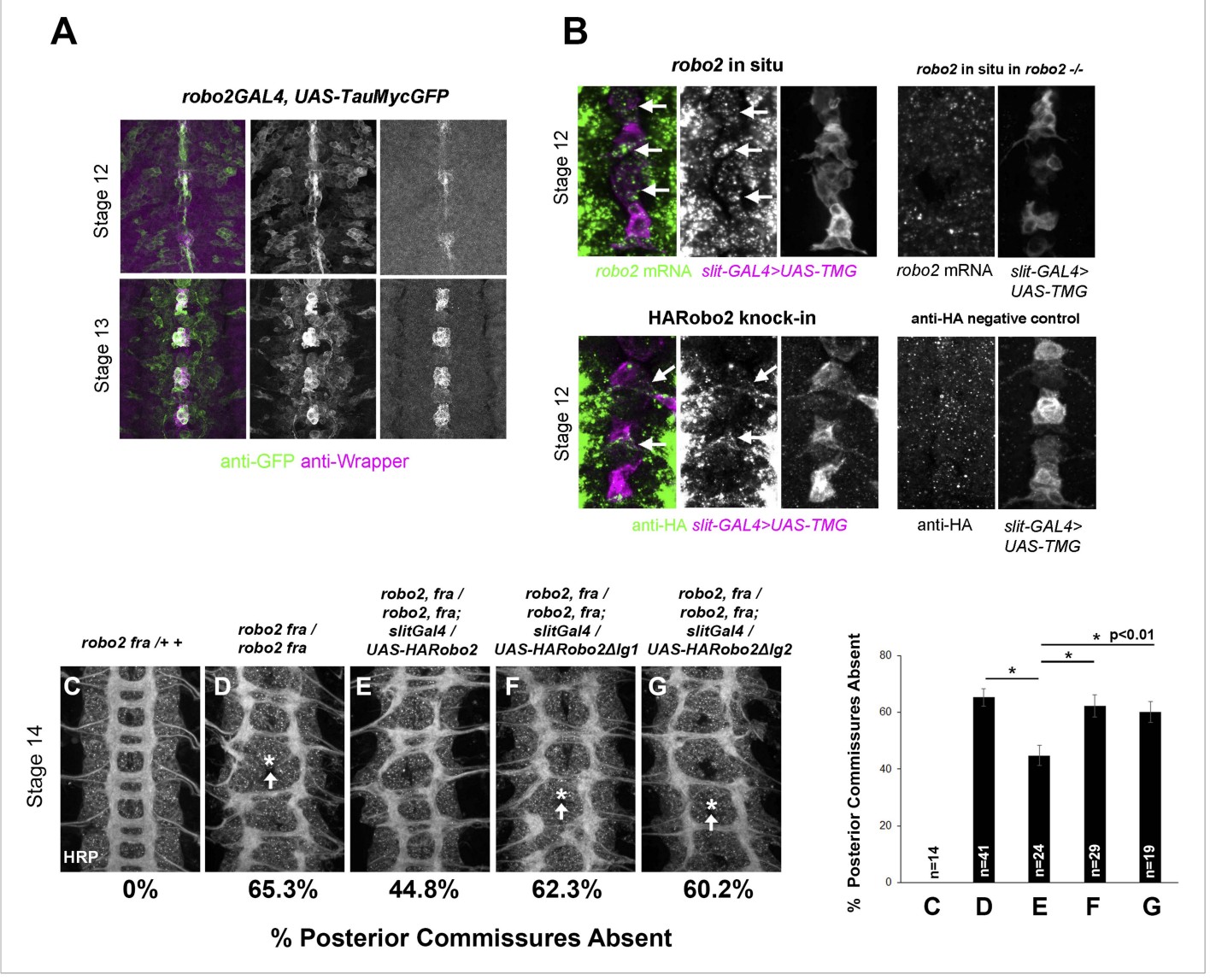

**Figure 8**. *robo2* is expressed in midline cells during commissural axon path finding, and over-expressing robo2 with *slit-GAL4* restores midline crossing in *robo2, fra* double mutants. (**A**) A *robo2-GAL4* enhancer trap that recapitulates *robo2*'s endogenous expression pattern drives *UAS-TauMycGFP* expression (green) in midline cells at stages 12–13, when many commissural axons cross the midline. Midline glia are labeled by an anti-wrapper antibody (magenta). (**B**) Top: Fluorescent in situ for *robo2* mRNA (green). *robo2* is transiently expressed in midline glia and neurons (magenta) during stage 12 (arrows). This in situ signal is not observed in *robo2* mutant embryos confirming the specificity of the mRNA detection (right). (**B**) Bottom: Robo2 protein is expressed in midline cells during the stages of commissural axon path finding, as shown by the expression pattern of a HA-tagged *robo2* cDNA knock-in allele (robo2^HArobo2). Stage 12 embryos carrying robo2^HArobo2, *slit-GAL4*, and *UAS-TauMycGFP* show HARobo2 expression in *slitGAL4*-expressing cells (arrows), whereas this signal is not detected in control embryos (right). (**C–G**) Stage 14 embryos of the indicated genotypes stained with anti-HRP antibodies to label all CNS axons. The absence of PC was scored in abdominal segments A1–A8 (arrows indicate examples of missing commissures). The PC defects of *robo2, fra* double mutants (**D**) are significantly rescued by over-expressing *UAS-Robo2* with *slit-GAL4* (**E**), whereas over-expression of *UAS-Robo2ΔIg1* (**F**) or *UAS-Robo2ΔIg2* (**G**) has no effect. Embryos were scored blind to genotype. Significance was assessed by one-way ANOVA followed by multiple comparisons using the Student's *t*-test and a Bonferonni correction (*$p < 0.01$). Error bars represent s.e.m. *n*, number of embryos scored for each genotype.

The following figure supplement is available for figure 8:

**Figure supplement 1**. *robo2* mRNA is transiently expressed in midline cells.

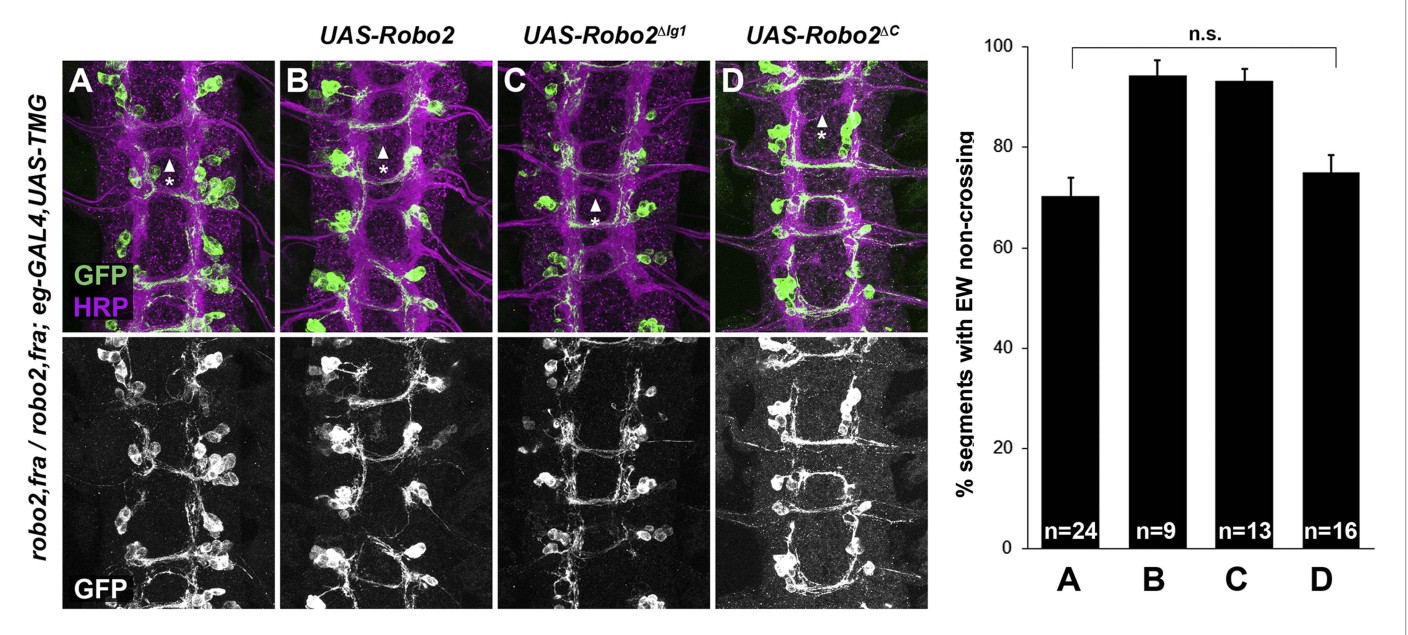

**Figure 9**. Robo2 cannot rescue midline crossing cell autonomously. (**A–D**) Stage 16 embryos of the indicated genotypes stained with anti-HRP (magenta) and anti-GFP (green) antibodies. Anti-GFP labels the EG and EW cell bodies and axons. EW crossing defects in *robo2, fra* double mutants (**A**) are not rescued by *eg-GAL4* mediated over-expression of *UAS-Robo2* (**B**), *UAS-Robo2ΔIG1* (**C**), or *UAS-Robo2ΔC* (**D**), suggesting that Robo2 cannot act cell autonomously to promote midline crossing. Segments with non-crossing EW axons are indicated by arrowheads with asterisks. Significance was assessed by multiple comparisons using the Student's *t*-test and a Bonferonni correction. No significant differences between any of the genotypes were observed (p > 0.3). Error bars represent s.e.m. *n*, number of embryos scored for each genotype.

plasma membrane in pre-crossing commissural axons, triggering premature Slit repulsion and preventing commissure formation. We reasoned that if Robo2 antagonizes Slit-Robo1 repulsion, then Robo2 mis-expression might restore midline crossing in *comm* mutant embryos. Indeed, pan-neural mis-expression of Robo2 with *elav-GAL4* significantly restored commissure formation in *comm* mutant embryos (*Figure 10*). To specifically test the Ig1/Ig2-dependence of Robo2's pro-crossing activity in this assay, we mis-expressed our Ig1+2 chimeric receptors (Robo1$^{R2Ig1+2}$ and Robo2$^{R1Ig1+2}$) with *elav-GAL4* in *comm* mutant embryos. We found that pan-neural mis-expression of Robo1$^{R2Ig1+2}$ in *comm* mutant embryos strongly suppressed the commissureless phenotype and restored midline crossing of many axons, as assayed by anti-HRP antibody staining, while mis-expression of Robo2$^{R1Ig1+2}$ had a much milder effect (*Figure 10*). These results suggest that Robo2 promotes midline crossing in an Ig1/Ig2-dependent manner by antagonizing canonical Slit-Robo1 repulsion.

We were also able to suppress the *comm* mutant phenotype by over-expressing Robo2 using *slit-GAL4*, and this effect was fully dependent on both Ig1 and Ig2 of Robo2 (*Figure 10*). This is consistent with our observations that Robo2 can act non-cell autonomously to promote ectopic midline crossing (*Figure 7*) and rescue midline crossing defects (*Figure 8*) in an Ig1/2-dependent manner. Of note, the suppressive effect of Robo2 expression in *comm* mutants is much greater when expressed in midline cells than when expressed pan-neurally (*Figure 10*). This is likely because when expressed pan-neurally, in addition to its pro-crossing activity, full-length Robo2 also has repulsive activity. In contrast, when expressed in midline cells, Robo2 would be unable to act as a repulsive receptor.

### Robo2 binds to Robo1 in vivo and the interaction depends on Ig1 and Ig2

As shown above, Robo2 is able to antagonize Slit-Robo1 repulsion in an Ig1/Ig2-dependent manner. One possibility is that Robo2 may form an inhibitory receptor–receptor complex with Robo1 to prevent it from signaling midline repulsion in response to Slit. If this is the case, we reasoned that we might be able to detect a physical interaction between Robo2 and Robo1 in embryonic protein

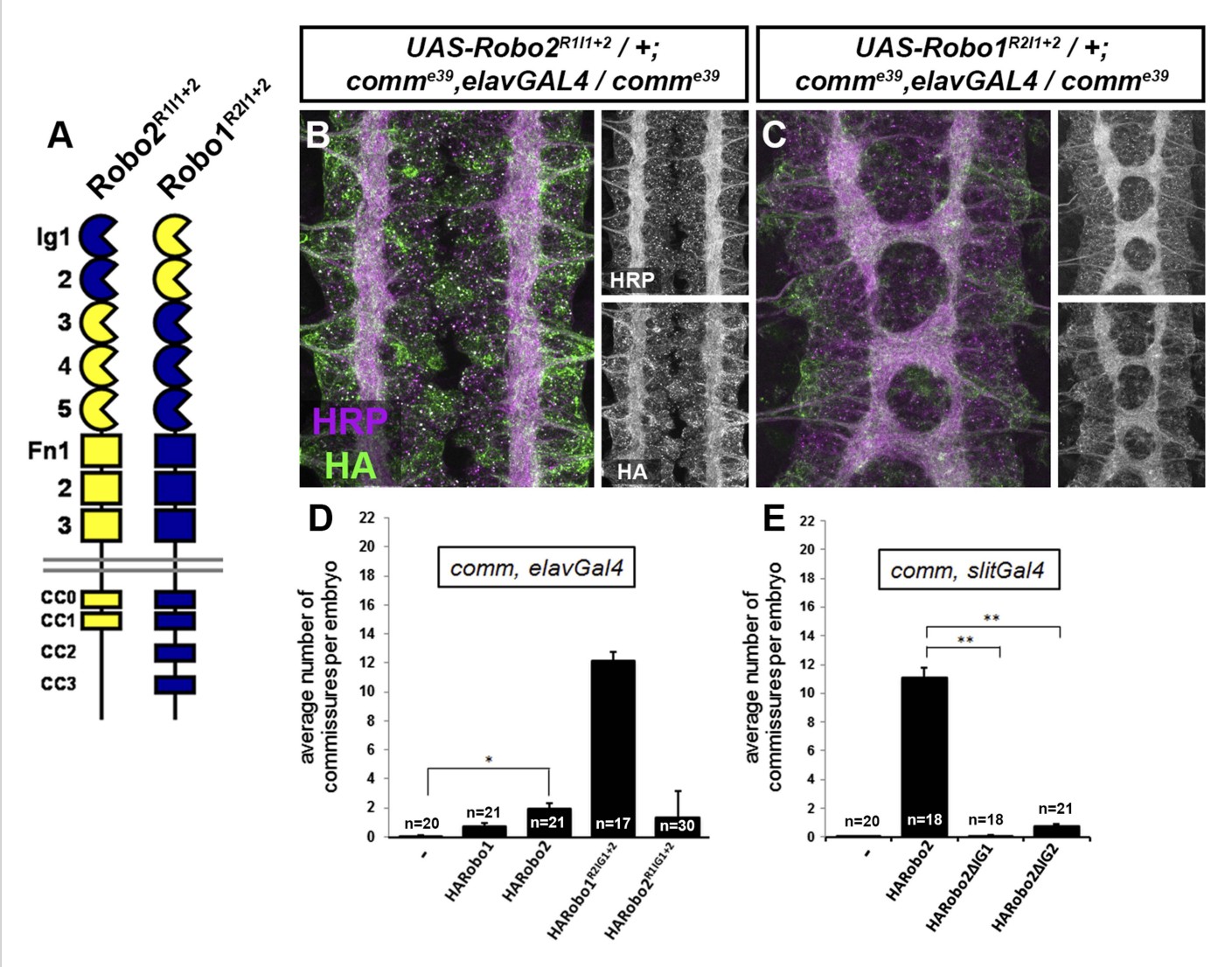

**Figure 10**. Robo2 receptors that promote midline crossing suppress *comm* mutants. (**A**) Schematic diagram of the two chimeric receptors shown in (**B** and **C**). Robo1 sequences are depicted in blue and Robo2 sequences are depicted in yellow. (**B** and **C**) Stage 16 embryos of the indicated genotype stained with anti-HRP to visualize CNS axons and anti-HA to visualize the epitope tagged chimeric receptor. Single channel images of HRP and HA are presented to the right of the color panels. Expression of the HA-Robo2$^{R1Ig1-2}$ chimeric receptor in a *comm* mutant background (**B**) does not restore commissure formation, while expression of the reciprocal HA-Robo1$^{R2Ig1+2}$ chimeric receptor (**C**) strongly suppresses the *comm* mutant phenotype. (**D** and **E**) Quantification of the average number of commissures per embryo in *comm* mutants expressing the indicated HA-tagged receptor transgenes in either all neurons using *elav-GAL4* (**D**) or in midline cells using *slit-GAL4* (**E**). Significance was assessed by one-way ANOVA followed by multiple comparisons using the Student's *t*-test and a Bonferonni correction (*p < 0.0001) (**p < 1.0 e−10). Error bars represent s.e.m. *n*, number of embryos scored for each genotype.

extracts. To test this idea, we mis-expressed epitope-tagged forms of Robo1 and Robo2 in *Drosophila* embryonic neurons with *elav-GAL4* and looked for physical interactions by co-immunoprecipitation (*Figure 11*). We found that Robo1-myc and HA-Robo2 co-immunoprecipitated from embryonic lysates when both were expressed in embryonic neurons (*Figure 11A*). Interactions were also observed between Robo1 and the closely related Robo3 receptor but not with a similarly tagged and structurally related Fra receptor (*Figure 11A*). As we would predict from our gain of function experiments, Robo2's ability to bind to Robo1 is independent of its cytoplasmic domain (*Figure 11*, *Figure 11—figure supplement 1*). Importantly, Robo2's ability to bind Robo1 depends on the Ig1–Ig2 region of Robo2, as Robo1$^{R2Ig1+2}$ was readily co-immunoprecipitated with Robo1, while

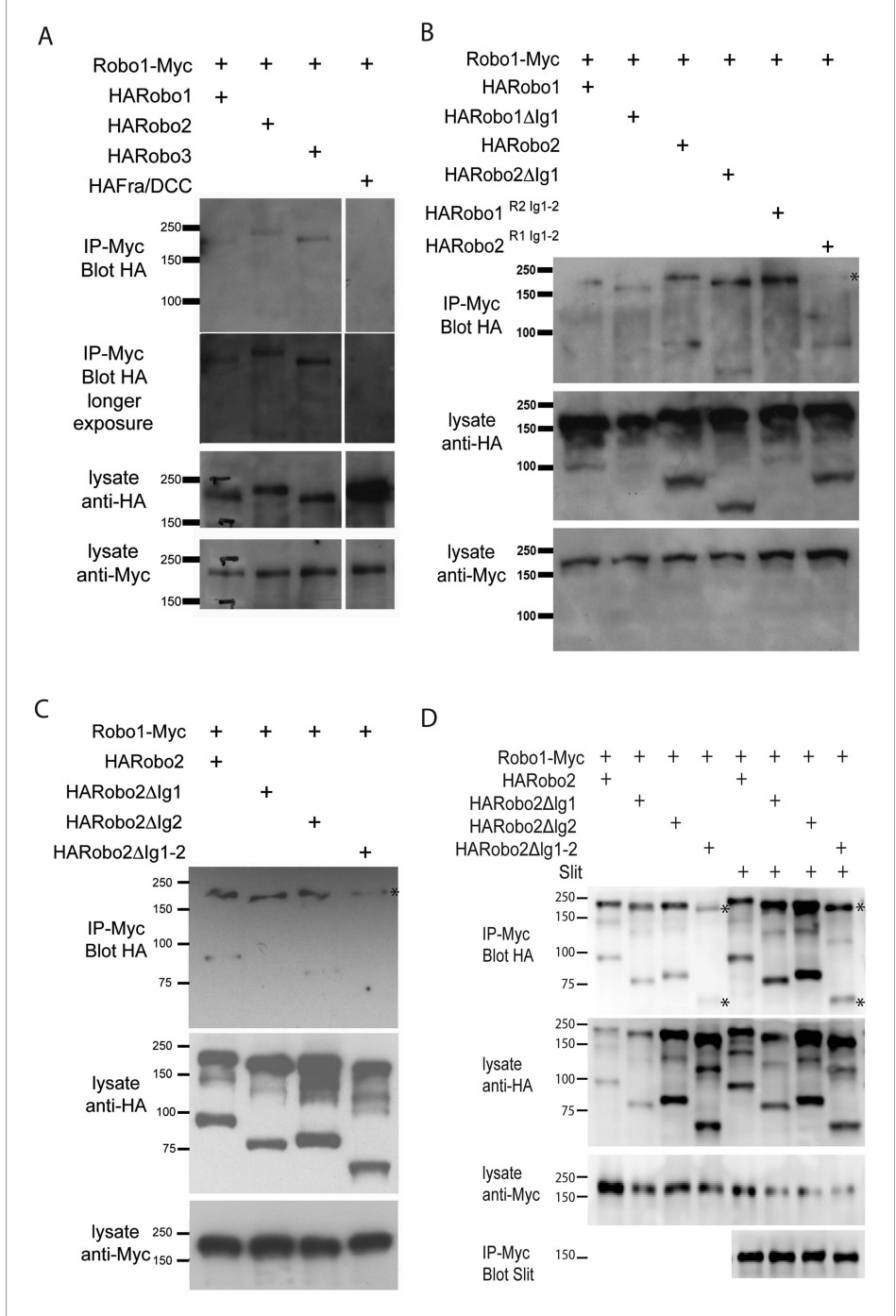

**Figure 11**. Robo2 binds to the Robo1 receptor in vitro and in vivo. (**A–C**) Protein extracts from embryos expressing Robo1-Myc and various HA-tagged receptors in all neurons were immunoprecipitated with anti-Myc antibodies and analyzed by western blot. Immunoprecipitates were probed with anti-HA (top blots) and total lysates were compared for HA expression and Myc expression to ensure that equal inputs were analyzed. Representative western blots from multiple experiments are shown. (**A**) Robo1-Myc binds to HARobo1, HARobo2, and HARobo3 but not to a HA-tagged Fra receptor (two exposures are shown). Total lysate blots reveal comparable loading with the exception of the Fra negative control in which there is substantially more HA-tagged receptor. (**B**) Robo1-Myc binds efficiently to HARobo2, HARobo2ΔIg1, and the HARobo1Robo2(IG1-2) chimera but not to the reciprocal chimera that has Ig1 and Ig2 domains from Robo1 (asterisk). (**C**) Deletion of either Robo2 Ig1 or Ig2 alone does not substantially affect Robo1 binding, while deleting both domains results in reduced binding (asterisk).
*Figure 11. continued on next page*

*Figure 11. Continued*

See *Figure 11—figure supplement 1* for an additional example. (**D**) Cell lysates of S2R+ cells separately transfected for Robo1-Myc or HA-tagged Robo2 variants were mixed, immunoprecipitated with anti-Myc, and analyzed by western blot. In this assay, Robo1-Myc binds efficiently to HARobo2, HARobo2ΔIg1, and HARobo2ΔIg2, and less well to HARobo2ΔIg1+2 (asterisks). In lanes 1–4, cells were untreated; in lanes 5–8, cells were treated with Slit-conditioned media before lysing. We note that in addition to detection of the predicted full-length Robo2 receptor with anti-HA, we also routinely detect a smaller ~80 kD fragment that corresponds to an extracellular domain cleavage product. The size of this fragment is shifted to predictably smaller sizes when Ig1, Ig2, or both Ig1 and Ig2 are deleted. We do not currently know, whether this cleavage event is required for Robo2 function in any context, since we have not been able to generate an uncleavable version of the receptor.

The following figure supplement is available for figure 11:

**Figure supplement 1**. Robo2 binding to Robo1 does not depend on its cytoplasmic domain or on Robo1's Ig1 domain.

binding between Robo1 and the reciprocal receptor Robo2$^{R1Ig1+2}$ was only weakly detected (*Figure 11B*). Consistently, deleting both of the Ig1 and Ig2 domains from Robo2 results in a diminished interaction with Robo1 in vivo and in vitro (*Figure 11*, *Figure 11—figure supplement 1*). The fact that not all binding is eliminated when both Ig1 and Ig2 are deleted suggests that other regions of the Robo2 receptor may contribute to the interaction with Robo1. Thus, we see a correlation between the presence of the Ig1 and Ig2 domains, a biochemical interaction with Robo1, and pro-crossing activity in the Robo2 receptor. These observations suggest that Robo2 may promote midline crossing through inhibitory interactions with Robo1, likely mediated at least in part by the Robo2 Ig1 and Ig2 domain. Of note, the Ig2 domain is essential for Robo2's pro-crossing activity but is not required for the interaction with Robo1 or Slit, suggesting the existence of an Ig2-specific activity that is distinct from the ability to bind Robo1 or Slit. One possible mechanism that could explain how receptor–receptor interactions could prevent Robo1 signaling is through blocking the access of Slit to the Ig1 region of Robo1. Deleting Robo1's Ig1 domain does not significantly attenuate the interaction with Robo2, but further experiments will be necessary to determine if Robo2 interferes with Robo1's interaction with Slit (*Figure 11*, *Figure 11—figure supplement 1*).

Our biochemical experiments examining receptor–receptor interactions when the Robo receptors are expressed in all neurons do not distinguish between cis and trans interactions. As we observed that Robo2 is able to non-cell autonomously inhibit Robo1 repulsion and promote midline crossing, we reasoned that we might be able to detect physical interactions between Robo1 and Robo2 receptors when they are presented in trans. We tested this prediction by transfecting *Drosophila* cultured S2R+ cells with either Robo1-myc or HA-tagged Robo2 and assaying for physical interactions by co-immunoprecipitation. Although we detected strong interactions between Robo1 and Robo2 in co-transfected cells, we could not detect interactions in cells that were transfected separately and mixed together (*Figure 11*, *Figure 11—figure supplement 1* and data not shown). However, when we mixed the membrane lysates of cells that were transfected separately, we observed that Robo1 readily co-immunoprecipitated Robo2, in an Ig1-2 dependent manner (*Figure 11*). These data suggest that physical interactions can occur between Robo1 and Robo2 receptors that are expressed in different cells and are consistent with the possibility of a physical interaction occurring across cell membranes in vivo. It is important to recognize that binding detected with mixed cell lysates could occur in either cis or trans. Future work should more rigorously evaluate the potential for trans interactions.

## Robo2's Ig2 domain is required for its endogenous activity in promoting midline crossing

Our data are consistent with a non-autonomous requirement for Ig1 and Ig2 of Robo2 in antagonizing Robo1 to promote midline crossing. However, the genetic data supporting this model arise from gain of function and rescue experiments using GAL4/UAS over-expression. In order to more rigorously address the endogenous requirement for Robo2 in promoting midline crossing, we generated modified BACs and evaluated the ability of either wild-type Robo2 or Robo2ΔIg2 to restore midline

crossing in *robo2, fra* double mutants, when expressed under *robo2*'s endogenous control elements. As Ig1 is required for both Robo2's pro-crossing activity and for its repulsive signaling output, the Robo2ΔIg2 variant provides a more specific reagent for testing our model. Therefore, we modified the original Robo2 BAC by recombineering to insert wild-type Robo2 cDNA or Robo2ΔIg2 cDNA, and introduced these BAC transgenes into *robo2, fra* double mutants. We determined the rescuing activity of each BAC through two assays: first, by scoring midline crossing of EW axons labeled by *eg-GAL4*, and second, by analyzing commissure formation in embryos stained with anti-HRP to label all axons (*Figure 12*).

We found that the ability of the Robo2 BAC to rescue midline crossing defects in *robo2, fra* double mutants was strongly impaired by deleting the Ig2 domain. In the EW crossing assay, one copy of the Robo2 FL cDNA BAC provides a significant rescue of *robo2, fra* double mutants at stage 16, whereas one copy of the Robo2ΔIg2 BAC has no effect (*Figure 12A–D*). Of note, removing one allele of *robo2* enhances midline crossing defects in *fra* mutants, explaining in part the incomplete rescue (*Figure 12*). In addition, it is likely that the Robo2 BAC does not contain all of the regulatory elements required for

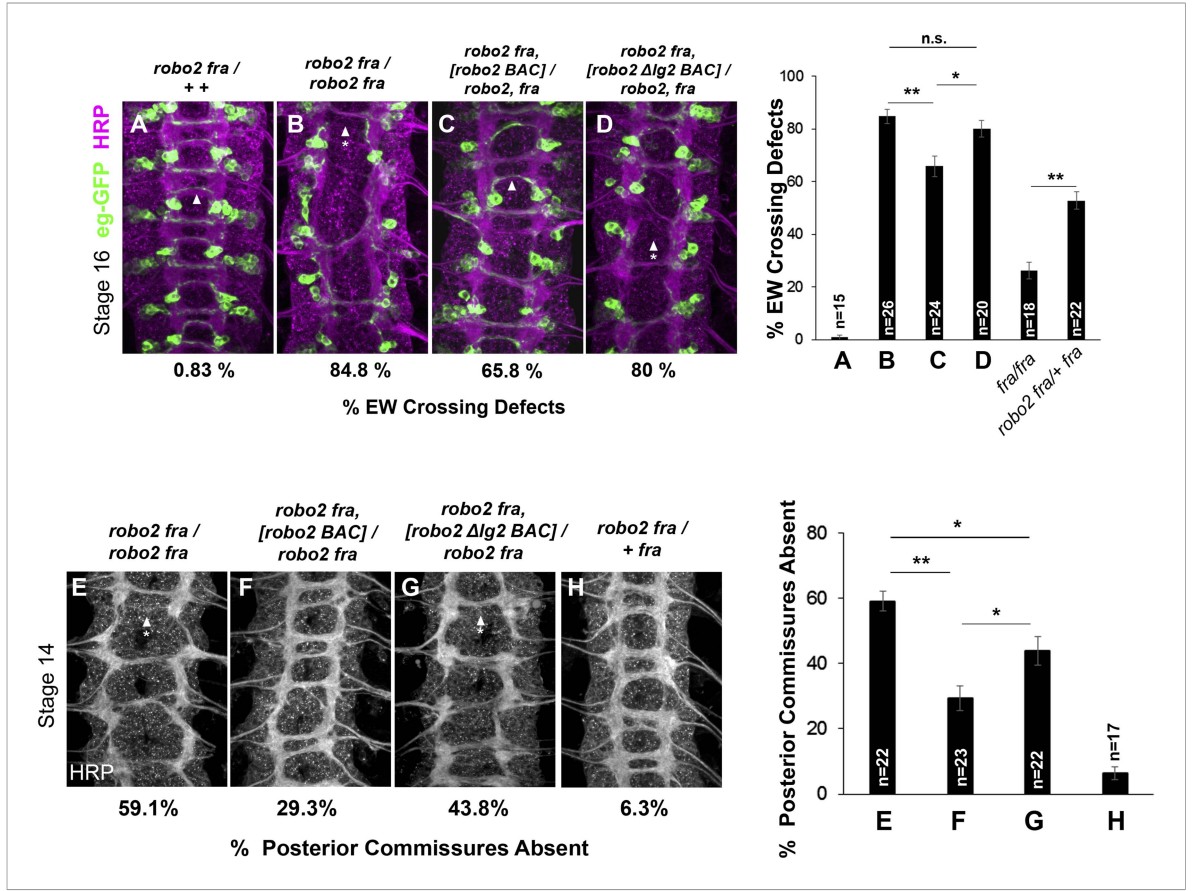

**Figure 12**. Robo2's endogenous activity in promoting midline crossing depends on Ig2. (A–D) Stage 16 embryos stained with anti-HRP (magenta) and anti-GFP (green) antibodies. Anti-GFP labels the EG and EW cell bodies and axons. Arrowheads indicate EW axons that have crossed the midline and arrowheads with asterisks indicate non-crossing EW axons. (A) Almost all EW axons cross the midline in *robo2, fra/+, +* double heterozygotes. (B) EW crossing defects are observed in 85% of segments in *robo2, fra* double mutants. (C–D) The FL Robo2 cDNA BAC transgene (C) significantly rescues EW crossing, to 66% of segments with defects (Student's *t*-test, **p < 0.001) whereas the Robo2ΔIG2 transgene (D) does not significantly rescue. Right: Removing one copy of *robo2* significantly enhances midline crossing defects in *fra* mutants. (E–H) Stage 14 embryos of the indicated genotypes stained with anti-HRP. Posterior commissures were scored in abdominal segments A1–A8. Missing posterior commissures are indicated by arrowheads with asterisks. (E–G) The posterior commissure defects of *robo2, fra* double mutants are significantly rescued by a full-length (FL) Robo2 cDNA BAC transgene (Student's *t*-test, **p < 0.001) (F), as well as by a Robo2ΔIG2 BAC (*p < 0.0167) (G). The Robo2ΔIG2 BAC does not rescue as well as FL Robo2 (*p < 0.0167). All embryos were scored blind to genotype. Significance was assessed by one-way ANOVA followed by multiple comparisons using the Student's *t*-test and a Bonferonni correction. Error bars represent s.e.m. *n*, number of embryos scored for each genotype.

*robo2*'s pro-crossing function, as one copy of the BAC does not restore EW crossing back to the levels of *fra* mutants heterozygous for *robo2* (*Figure 12*).

We also assessed the ability of the BAC transgenes to rescue midline crossing defects when analyzing all axons using anti-HRP. By this method, we see a robust rescue in posterior commissure (PC) formation in *robo2, fra* double mutant embryos with one copy of the Robo2 cDNA BAC compared to controls (*Figure 12E–H*). In contrast, the Robo2ΔIg2 BAC provides a much weaker rescue (*Figure 12G*). The partial rescue by the Robo2ΔIg2 BAC in this assay suggests that the severe *fra, robo2* phenotype is due to the combined requirement for multiple activities of Robo2, including one that is Ig2-independent. Nevertheless, these data unambiguously reveal an endogenous requirement for Robo2's Ig2 domain during commissural axon guidance. Importantly, the Robo2ΔIg2 BAC fully rescues Robo2's repulsive activity at the midline (data not shown), further demonstrating that the Ig2 domain is specifically required for Robo2 to successfully promote midline crossing, but not for other known activities of the Robo2 receptor. Taken together, these results demonstrate a requirement for Robo2's Ig2 domain in promoting midline crossing when expressed under its endogenous control elements and strongly support the model that Robo2 promotes midline crossing of commissural axons by antagonizing repulsion through an Ig1/Ig2-mediated inhibitory interaction with Robo1.

## Discussion

In this manuscript, we have described a role for Robo2 in promoting midline crossing through inhibition of Slit-Robo1 repulsion. Loss of function experiments point to an endogenous requirement for Robo2 in promoting midline crossing. Additional genetic analyses indicate that Robo2 can antagonize Robo1 in the absence of its cytoplasmic domain and that this inhibitory effect can be generated by non-cell autonomous expression of Robo2. These observations, together with the demonstration that Robo2 variants that promote midline crossing are potent suppressors of *comm* mutants, support the model that Robo2 inhibits Slit-Robo1 repulsion, rather than acting as a receptor that promotes midline axon attraction. Biochemical and gain of function genetic analyses show that Robo2 can bind to Robo1 in vivo through its Ig1 and Ig2 domains and that this binding interaction correlates with Robo2's pro-crossing activity. Furthermore, cell type specific rescue experiments and analysis of Robo2 mRNA and protein expression are consistent with a requirement for Robo2 in midline cells and support an endogenous requirement for Robo2's Ig2 domain in promoting midline crossing. Taken together, the data in this manuscript support the model that Robo2 expressed in cells other than commissural neurons acts to inhibit Robo1 receptor activity through extracellular domain binding interactions, and that this activity ensures the precise execution of midline guidance (*Figure 13*). Our model reconciles two previously confounding observations: one, that a small amount of Robo1 protein is detectable on commissural axons as they cross the midline, yet this pool of Robo1 is unable to signal midline repulsion; and two, that the single known isoform of Robo2 can act to both promote and inhibit midline crossing.

### Multiple mechanisms ensure precise and robust regulation of Robo1 repulsion

Given the prominent role that Comm plays in regulating Robo1 receptor expression to prevent premature responses to Slit, it is fair to ask why it is necessary to invoke a second mechanism to down-regulate Robo1 receptor signaling. Indeed, in wild-type animals, there is no obvious requirement for Robo2's pro-crossing activity, at least not at the embryonic midline in the populations of neurons that we have assayed. A requirement for Robo2 in promoting midline crossing in otherwise wild-type animals has been described for the guidance of foreleg gustatory neurons in the adult nervous system, although it is not clear in this context if the same mechanism that we have described is at work (*Mellert et al., 2010*). Nevertheless, a clear endogenous contribution for Robo2 at the embryonic midline can be demonstrated in conditions where attractive guidance cues, such as Netrin, are compromised. One probable explanation for the existence of this second regulatory mechanism is that it confers robustness on the essential process of midline circuit formation, and that this is important to the animal when developmental conditions are not optimal.

While Comm is an efficient and potent negative regulator of Robo1 trafficking to the growth cone surface, it is clear that not all Robo1 is prevented from reaching the surface in the presence of Comm.

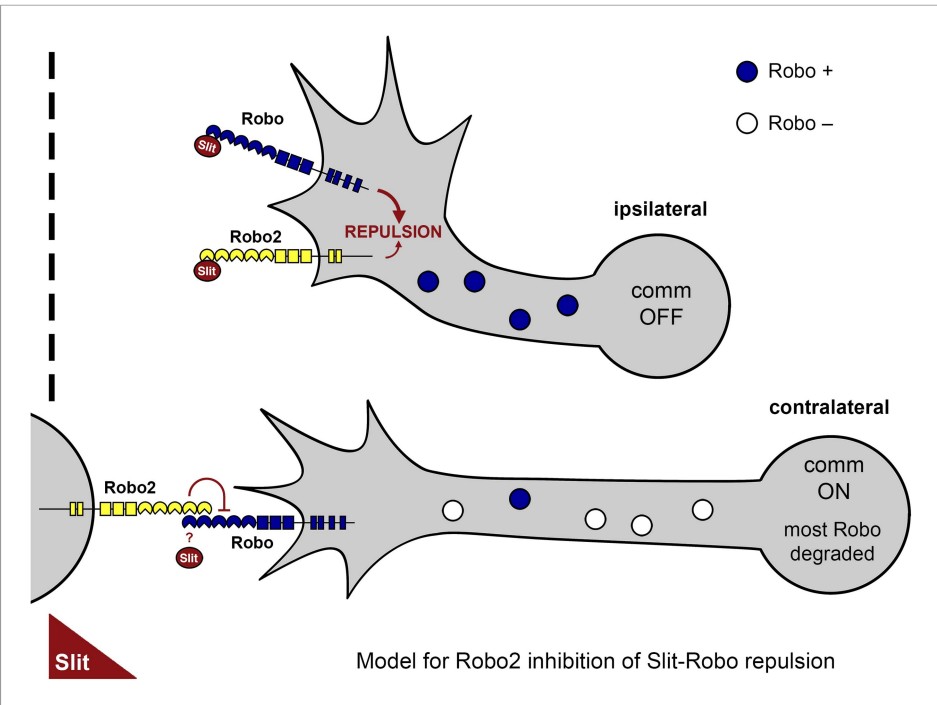

**Figure 13**. A model of Robo2's pro-crossing function. In contralateral neurons, the endosomal sorting receptor Comm is expressed, and it prevents the majority of Robo1 from reaching the growth cone surface. We propose that Robo2 acts non-autonomously in midline cells to bind to and inhibit the low level of Robo1 that escapes Comm-dependent sorting. This mechanism is revealed in contexts where axon attraction to the midline is limited. In ipsilateral neurons, Comm is not expressed, and Robo2 works together with Robo1 to mediate repulsion from the midline in response to Slit.

Low levels of Robo1 can be detected on commissural axons by immunostaining and immunoelectron microscopy (*Kidd et al., 1998a*). Data from surface labeling experiments indicate that Comm acts on newly synthesized Robo1, and the question of how Robo1 receptors already present on the plasma membrane prior to the initiation of *comm* expression might be regulated remains unresolved (*Keleman et al., 2002*). The role of Robo2 may thus be to negatively regulate the low levels of Robo1 that escape Comm-dependent sorting.

In addition to the complementary actions of Comm, a cell autonomous regulator of Robo1 trafficking (*Keleman et al., 2002*, *2005*), and Robo2, a cell non-autonomous inhibitor of Robo1 signaling (this study), it is likely that there are additional levels of regulation that contribute to preventing premature response to midline Slit. In particular, a recent study shows quite convincingly that Comm's role in sorting Robo1 is insufficient to explain how Robo1 activity is limited in pre-crossing commissural axons. Specifically, embryos in which the endogenous Robo1 receptor is replaced with a variant of Robo1 that is insensitive to the sorting activity of Comm by homologous recombination show no defects in midline crossing (*Gilestro, 2008*). This observation is in marked contrast to the prediction of the sorting model, in which embryos carrying a Comm-resistant Robo1 receptor would be expected to resemble *comm* mutants. It will be of great interest to obtain an explanation for this paradoxical finding and to determine what additional targets of Comm could also regulate Slit-dependent repulsion.

## Inhibitory receptor–receptor interactions in trans: a new mechanism to regulate axon guidance

Our results suggest that Robo2 can inhibit Robo1 activity and that this effect is mediated by receptor–receptor interactions between the Robo2 and Robo1 extracellular domains. Cis-inhibitory interactions, such as those that occur between the transmembrane protein Kekkon 1 and the

epidermal growth factor receptor (EGFr) (*Ghiglione et al., 2003*), and between ligand and receptor pairs, as in the cases of Ephs/ephrins and Notch/Delta, have been well documented (*del Alamo et al., 2011*; *Kao and Kania, 2011*; *Yaron and Sprinzak, 2012*). While we were not able to detect trans interactions by co-immunoprecipitation or by an S2 cell aggregation assay (data not shown), our genetic data strongly suggest that Robo2 acts in trans to inhibit Robo1 signaling. A recent in vitro screen for trans interactions among *Drosophila* cell surface receptors did not report a direct interaction between Robo1 and Robo2, suggesting that if trans interactions do occur, they might be mediated by a cofactor (*Ökzan et al., 2013*). Indeed, Slit-dependent trans interactions between Robo1 and Robo2 have been proposed to play a role in the migration of sensory neurons in the *Drosophila* peripheral nervous system, although in this case Robo2 is thought to promote Slit-Robo repulsive signaling by presenting Slit to Robo receptors expressed in trans (*Kraut and Zinn, 2004*).

Previous studies have defined growth factor and morphogen receptor regulatory mechanisms that bear some resemblance to the mechanism that we have described here. For example, EGFr signaling and Bone Morphogenetic Protein receptor (BMPr) signaling can be attenuated cell non-autonomously by various inhibitory factors, such as Argos for EGFrs and Noggin for BMPrs (*Klein et al., 2004*; *Walsh et al., 2010*). In the case of EGFr, receptor signaling is blocked because the soluble inhibitory factor Argos binds to and sequesters the EGF ligand, thereby preventing receptor activation (*Klein et al., 2008*). The mechanism through which Robo2 regulates Robo1 is similar in that it acts cell non-autonomously and that it depends on extracellular interactions, but distinct, since Robo2 does not appear to act solely by binding and sequestering Slit, as the Robo2ΔIg2 receptor can still bind Slit, but is completely unable to inhibit Robo1 activity.

It remains to be determined, but a closer analogy may exist with the way Dickkopf (DKK) family proteins antagonize Wnt receptor signaling (*Niehrs, 2006*). In this case, secreted DKK binds to the lipoprotein related proteins (LRP5 and 6), which are co-receptors for Wnt, and prevents LRP interaction with the Frizzled/Wnt ligand receptor complex (*Ahn et al., 2011*; *Chen et al., 2011*). While Robo2 is not secreted, there is evidence that Robo1 receptor extracellular domains can be cleaved and shed into the extracellular space (*Coleman et al., 2010*), and we have observed that the Robo2 ectodomain can also be shed in vitro and in vivo (Evans and Bashaw, unpublished). In the future, it will be interesting to investigate whether Robo2 binding prevents Robo1 from interacting with Slit in vivo, and whether Robo2 receptor cleavage is important for its ability to promote midline crossing. Alternatively, Robo2 could prevent the recruitment of Robo1's downstream signaling molecules such as Enabled, Nck/Dock and Son of Sevenless (*Bashaw et al., 2000*; *Fan et al., 2003*; *Yang and Bashaw, 2006*).

## How are the diverse axon guidance activities of the Robo2 receptor coordinated?

In addition to its Ig1/Ig2 dependent role in inhibiting Slit-Robo1 repulsion that we have described here, Robo2 has at least three other distinct axon guidance activities that can be attributed to different structural elements of the receptor. In the context of midline axon repulsion, Robo2 binds Slit through its extracellular Ig1 domain and cooperates with Robo1 to prevent abnormal midline crossing. It is not known how Robo2 signals repulsion, but based on receptor swap experiments that demonstrate that Robo1 can substitute for Robo2's midline repulsive activity, it seems likely that a common cytoplasmic signaling output shared by Robo1 and Robo2 (perhaps mediated by the shared CC0 or CC1 motifs) is important for repulsion (*Spitzweck et al., 2010*). Robo2 also directs the mediolateral position of axons in the CNS, an activity conferred by a combination of its extracellular Ig1 and Ig3 domains (*Evans and Bashaw, 2010b*). In this context, distinct biochemical properties conferred by Ig3 appear to direct Robo2 receptor multimerization, and this property correlates with the ability to regulate lateral position in vivo (*Evans and Bashaw, 2010b*). Finally, in addition to these activities, we have recently discovered a new function for Robo2 in regulating the guidance of specific populations of motor axons to their appropriate muscle targets. In this case, Robo2's guidance activity depends on unique features of its cytoplasmic domain (*Santiago et al., 2014*).

A major challenge for the future will be to understand how these diverse guidance activities are selectively deployed at the right time and place to allow for coordinated guidance responses. One important factor that is likely to contribute to the coordination of these activities is the regulation of the spatial and temporal expression of Robo2. For example, in late stage embryos, Robo2 protein

expression is restricted to the lateral most regions of the longitudinal connectives where it is presumably acting to control lateral positioning, while in younger embryos *robo2* mRNA can be detected in ipsilateral pioneer neurons where it is likely contributing to midline repulsion. Robo2 is also detected in midline glia and neurons, where we propose it may act to prevent premature responses to Slit. At present, little is known about how these patterns of expression are established and temporally regulated, although we have recently shown that the homeodomain transcription factors *dHb9* and *nkx6* are required for *robo2* expression in a subset of motor neurons (*Santiago et al., 2014*).

While controlling the time and place of Robo2 expression is no doubt part of the explanation for how Robo2's diverse and sometimes opposing activities are coordinated, we expect that the distinct biochemical features of Robo2's different activities, as well as the potential interaction with context-specific cofactors will also play an important role. Here, we note that Robo2 does not appear to be able to promote midline crossing cell-autonomously, either in subsets of commissural neurons in rescue experiments, or in the apterous ipsilateral interneurons in gain-of-function experiments. This could be because Robo2 is unable to bind to Robo1 in cis in vivo, or alternatively because Robo1–Robo2 cis interactions confer a distinct outcome from the inhibitory effect of Robo2 presented from other cells. This is reminiscent of the different responses produced by cis and trans interactions between receptors and their ligands (*Yaron and Sprinzak, 2012*). How distinct signaling responses are triggered by the different structural conformations resulting from cis vs trans interactions remains poorly understood. Future experiments to define the mechanisms that control the specific expression domains and biochemical activities of Robo2 promise to continue to offer new insights into the molecular biology of axon guidance.

## Materials and methods

### Experimental Procedures

#### Genetics

The following *Drosophila* mutant alleles were used: *fra³*, *fra⁴*, *robo2ˣ³³*, *robo2ˣ¹²³*, *robo2ˣ¹³⁵*, *commᴱ³⁹*, *P{GawB}NP6273* (*robo2ᴳᴬᴸ⁴*), *egᴹᶻ³⁶⁰* (*eg-GAL4*), *ap-GAL4*, *dHb9ᴳᴬᴸ⁴*, *robo2ᴴᴬʳᵒᵇᵒ²*. The following transgenes were used: *P{10UAS-HARobo1}86Fb*, *P{10UAS-HARobo2}86Fb*, *P{10UAS-HARobo2ᐞᴵᵍ¹}86Fb*, *P{10UAS-HARobo2ᐞᴵᵍ²}86Fb*, *P{10UAS-HARobo2ᐞᴵᵍ¹⁺²}86Fb*, *P{10UAS-HARobo1ᐞᶜ}86Fb*, *P{10UAS-HARobo2ᐞᶜ}86Fb*, *P{UAS-HARobo1ᴿ²ᴵ¹⁺²}86Fb*, *P{UAS-HARobo2ᴿ¹ᴵ¹⁺²}86Fb*, *P{UAS-Robo1ᐞᶜmyc}2*, *P{UAS-Robo2ᐞᶜmyc}1*, *P{UAS-HARobo1ᴿ²ᴵ¹⁺²}*, *P{UAS-HARobo2ᴿ¹ᴵ¹⁺²}*, *P{UAS-Robo1myc}*, *P{UAS-HARobo2}T1*, *P{GAL4-elav.L}3* (*elav-GAL4*), *slit-GAL4*, *P{UAS-TauMycGFP}II*, *P{UAS-TauMycGFP}III*. Transgenic flies were generated by BestGene Inc. (Chino Hills, CA) or Rainbow Transgenic Flies Inc. (Camarillo, CA) using ΦC31-directed site-specific integration into landing sites at cytological position 86F (for UAS-Robo constructs) or 51C (for *robo2* BAC CH321-22K18 and modified BACs). All crosses were carried out at 25°C. Embryos were genotyped using balancer chromosomes carrying *lacZ* markers or by the presence of epitope-tagged transgenes.

#### Molecular biology

##### pUAST cloning

Robo coding sequences were cloned into a pUAST vector (p10UASTattB) including 10xUAS and an attB site for ΦC31-directed site-specific integration. All p10UASTattB constructs include identical heterologous 5′ UTR and signal sequences (derived from the *Drosophila* wingless gene) and an N-terminal 3×HA tag. Robo domain deletion variants created for this study were generated by PCR and include the following amino acids (numbers refer to Genbank reference sequences AAF46887 [Robo1] and AAF51375 [Robo2]): Robo2ᐞᴵᵍ¹ (187–1463), Robo2ᐞᴵᵍ² (84–186, 281–1463), Robo2ᐞᴵᵍ¹⁺² (281–1463), Robo1ᐞᶜ (56–950), Robo2ᐞᶜ (84–1022).

##### robo2 BAC and recombineering

The *robo2* BAC CH321-22K18 was generated by the P[acman] consortium (*Venken et al., 2009*) and obtained from BACPAC Resources (bacpac.chori.org). Modified BACs were generated by replacing *robo2* exons 2–14 and intervening introns with untagged or HA-tagged cDNAs via recombineering. Briefly, partial *robo2* cDNAs plus a kanamycin-resistance selective marker were cloned into a plasmid

vector flanked by 50 bp homology arms matching the 3′ end of the *robo2* first intron and the beginning of the *robo2* 3′ UTR. This cassette was excised by PmeI digestion and electroporated into DY380 cells containing the original CH321-22K18 BAC in which expression of lambda recombination genes had been induced by heat shock. Potential recombinant BACs were selected on LB plates containing chloramphenicol (12.5 µg/ml) and kanamycin (25 µg/ml), and verified by PCR amplification and sequencing of the entire recombineered region.

## Immunofluorescence and imaging

Dechorionated, formaldehyde-fixed, methanol-devittellinized embryos were fluorescently stained using standard methods. The following antibodies were used in this study: Rabbit anti-HA (Covance, Princeton NJ, PRB-101C, 1:2000), Mouse anti beta-tubulin (E7, DSHB, 1:100), Mouse anti-HA (Covance 16B12, 1:250), FITC-conjugated goat anti-HRP (Jackson Immunoresearch, West Grove PA, #123-095-021, 1:250), Alexa-647 conjugated goat-anti-HRP (Jackson #123-605-021, 1:500), mouse anti-Fasciclin-II/mAb 1D4 (Developmental Studies Hybridoma Bank, (DSHB, Iowa City IA, 1:100), mouse anti-βgal (DSHB, 1:150), rabbit anti-GFP (Invitrogen, Waltham MA, #A11122, 1:500), rabbit anti-c-Myc (Sigma, St. Louis MO, C3956, 1:500), Cyanine 3-conjugated goat anti-mouse (Jackson #115-165-003, 1:1000), Alexa-488-conjugated goat anti-rabbit (Molecular Probes, Eugene OR, #A11008, 1:500). Embryos were mounted in 70% glycerol/PBS. Fluorescent mRNA in situ hybridization was performed as described, with digoxigenin labeled probe (*Yang et al., 2009*). Briefly, hybridized probe was detected with anti-digoxigenin-HRP (Roche), using fluorescein-labeled tyramide as a substrate (TSA Fluorescence System, Perkin Elmer, Waltham MA). Phenotypes were analyzed and images were acquired using a spinning disk confocal system (Perkin Elmer) built on a Nikon Ti-U inverted microscope using a Nikon OFN25 60× objective with a Hamamatsu C10600-10B CCD camera and Yokogawa CSU-10 scanner head with Volocity imaging software. Images were processed using ImageJ. For fluorescence quantification of HA and Slit antibody staining in S2R+ cells, ImageJ was used to generate max projections and to obtain the average pixel intensity for a region of interest (four cells or more). For each channel, relative fluorescence intensity was calculated by dividing the average pixel intensity of transfected cells by that of untransfected cells from the same slide. The values from three images per Robo2 variant (Robo2, Robo2ΔIg1, Robo2ΔIg2, Robo2ΔIg1+2) were averaged; all images were acquired using identical imaging parameters.

For fluorescence quantification of HA antibody staining in embryos, three embryos per UAS line (Robo2, Robo2ΔIg1, Robo2ΔIg2, Robo2ΔIg1+2) were imaged using identical settings. Max projections were generated using ImageJ, and the average pixel intensity was measured across five regions within longitudinal axons for each embryo. The values from three embryos for each line were averaged.

## Biochemistry

### Slit binding assay

*Drosophila* S2R+ cells were cultured at 25°C in Schneider's media plus 10% fetal calf serum. To assay Slit binding, cells were plated on poly-L-lysine coated coverslips in six-well plates (Robo-expressing cells) or untreated six-well plates (Slit-expressing cells) at a density of 1–2 × 10⁶ cells/ml, and transfected with pRmHA3-GAL4 and HA-tagged pUAST-Robo or untagged pUAST-Slit plasmids using Effectene transfection reagent (Qiagen, Valencia CA). GAL4 expression was induced with 0.5 mM CuSO$_4$ for 24 hr, then Slit-conditioned media was harvested by adding heparin (2.5 µg/ml) to Slit-transfected cells and incubating at room temperature for 20 min with gentle agitation. Robo-transfected cells were incubated with Slit-conditioned media at room temperature for 20 min, then washed with PBS and fixed for 20 min at 4°C in 4% formaldehyde. Cells were permeabilized with PBS + 0.1% Triton X-100, then stained with antibodies diluted in PBS + 2 mg/ml BSA. Antibodies used were: mouse anti-SlitC (c555.6D, DSHB, 1:50), rabbit anti-HA (Covance, 1:2000), Cy3 goat anti-mouse (Jackson Immunoresearch, 1:500), and Alexa488 goat anti-rabbit (Molecular Probes, 1:500). After antibody staining, coverslips with cells attached were mounted in Aquamount. Confocal stacks were collected using a Leica SP5 confocal microscope and processed by NIH ImageJ and Adobe Photoshop software.

### Surface labeling

For surface labeling in S2R+ cells, cells were plated on poly-L-lysine coated coverslips and transfected with pRmHA3-GAL4 and HA-tagged pUAST-Robo2 plasmids using Effectene, as described above.

GAL4 expression was induced with 0.5 mM $CuSO_4$ for 24 hr, then cells were washed in cold PBS and blocked in PBS + 5% normal goat serum (NGS) for 20 min at 4°C. Cells were incubated in primary antibodies diluted in PBS + 5% NGS for 30 min at 4°C, then washed three times in cold PBS. Cells were fixed for 15 min at 4°C in 4% paraformaldehyde (PFA) in PBS, followed by three washes in PBS and incubation with secondary antibodies diluted in PBS + 5% NGS for 30 min at room temperature. For staining with detergent, cells were fixed 24 hr after GAL4 induction in 4% PFA for 15 min at room temperature, permeabilized in 0.1% Triton/PBS (PBT) for 5 min, blocked in PBT + 5% NGS for 20 min, and incubated overnight in primary antibodies diluted in PBT + 5% NGS. After three washes in PBT, secondary antibodies were added as described above. After secondary antibodies, cells were washed three times in PBS and coverslips were mounted in Aquamount. Antibodies used: mouse anti-beta tubulin (E7, DSHB, 1:100), rabbit anti-HA (Covance, 1:2000), Cy3 goat anti-rabbit (Jackson Immunoresearch, 1:2000), and Alexa488 goat anti-mouse (Molecular Probes, 1:1000).

## Co-immunoprecipitation

Approximately 100 µl of embryos co-expressing Myc-tagged and HA-tagged UAS Robo transgenes in all neurons with *elav-GAL4* were lysed in 0.5 ml of TBS-V (150 mM NaCl, 10 mM Tris ph8, 1 mM ortho-vanadate) supplemented with 1% Surfact-AMPS NP40 (Thermo, Waltham MA), protease inhibitors (Roche Complete), and 1 mM phenylmethanesulfonylfluoride (PMSF) by manual homogenization using a plastic pestle. After homogenization, embryos were incubated with gentle rocking at 4°C for 10 min and centrifuged in a pre-chilled rotor for 10 min at 14000 rpm. The soluble phase was removed and incubated with 1–2 µg of anti-Myc antibody (Millipore, Billerica MA) for 45 min with gentle rocking at 4°C. 50 µl of 50% slurry of proteinA and proteinG agarose (Invitrogen) were added to the tubes and samples were incubated for an additional 30 min with gentle rocking at 4°C. Samples were washed three times in lysis buffer and then boiled for 10 min in 50 µl of 2× Laemmli SDS Sample Buffer. Proteins were resolved by SDS Page and transferred to nitrocellulose for subsequent incubation with anti-myc (9E10, DHSB) 1:1000 or anti-HA (16B12 Covance) 1:1000 overnight at 4°C in PBS supplemented with 5% dry milk and 0.1% Tween 20. After three washes in PBS/0.1% Tween 20, HRP-conjugated secondary antibodies were applied for 1 hr at room temperature. Signals were detected using either ECL 2 or ECL Prime (Amersham, Amersham UK) according to manufacturers instructions.

For co-immunoprecipitation in *Drosophila* S2R+ cells, $10^6$ cells were transfected with pRmHA3-GAL4, HA or Myc-tagged pUAST-Robo or untagged pUAST-Slit plasmids and induced 24 hr after transfection, as described above. 48 hr after transfection, cells were lysed in TBS-V (150 mM NaCl, 10 mM Tris ph8, 1 mM ortho-vanadate) supplemented with 0.5% Surfact-AMPS NP40 (Thermo), protease inhibitors (Roche Complete), and 1 mM PMSF. Lysates were precleared with Protein A/G agarose for 30 min at 4°C, followed by addition of 1–2 µg of Rabbit anti-Myc (Millipore 06–549) or Rabbit anti-HA (Covance) for 1 hr at 4°C. 50 µl of 50% slurry of proteinA and proteinG agarose (Invitrogen) were added, and samples were incubated for an additional 30 min with gentle rocking at 4°C. Samples were washed 3× in lysis buffer and boiled for 10 min in 50 µl of 2× Laemmli SDS Sample Buffer.

For lysate mixing experiments, Slit-conditioned media was harvested 48 hr after transfection, as described above. pUAST-Robo1 and pUAST-Robo2 cell lysates were mixed for 1 hr at 4°C with gentle agitation before immunoprecipitation. In some conditions, Slit-conditioned media was added at 2× concentration to pUAST-Robo1 and pUAST-Robo2 cell lysates. SDS electrophoresis and Western blotting were performed as described above, and developed using WesternSure PREMIUM Chemiluminescent Substrate (Li-cor, Lincoln NE) according to manufacturer's instructions.

## Statistics

For statistical analysis, comparisons were made between genotypes using the Student's *t*-test. For multiple comparisons, significance was assessed by using a Bonferroni correction.

## Acknowledgements

We would like to thank members of the Bashaw lab for their thoughtful comments and ideas during the development of this manuscript. We thank Dr Barry Dickson for providing the Robo knock-in alleles.

## Additional information

### Funding

| Funder | Grant reference | Author |
| --- | --- | --- |
| National Institutes of Health (NIH) | RO1NS046333 | Greg J Bashaw |
| National Institutes of Health (NIH) | RO1NS054739 | Greg J Bashaw |
| National Institutes of Health (NIH) | F32-NS-060357 | Timothy A Evans |
| National Science Foundation (NSF) | DGE-0822 | Celine Santiago |

The funders had no role in study design, data collection and interpretation, or the decision to submit the work for publication.

### Author contributions

TAE, CS, GJB, Conception and design, Acquisition of data, Analysis and interpretation of data, Drafting or revising the article; EA, Acquisition of data, Analysis and interpretation of data

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
