## [Decision Letter]

Thank you for submitting your work entitled “Robo2 acts in trans to inhibit Slit-Robo1 repulsion in pre-crossing commissural axons” for peer review at *eLife*. Your submission has been favorably evaluated by K VijayRaghavan (Senior editor) and three reviewers, one of whom is a member of our Board of Reviewing Editors.

The reviewers have discussed the reviews with one another and the Reviewing editor has drafted this decision to help you prepare a revised submission. The following individuals responsible for the peer review of your submission have agreed to reveal their identity: Graeme Davis (Reviewing editor); Troy Littleton; and Esther Stoekli (peer reviewers).

Summary:

The manuscript by Evans et al. describes a role for *Drosophila* Robo2 in the promotion of midline crossing by embryonic CNS axons. The work builds upon an earlier finding from the Dickson lab that demonstrated an unexpected role for Robo2 in promoting midline crossing, in addition to its previously characterized role as a midline repellant. In the current work, the authors expand upon this finding and provide data that suggests Robo2 functions in promoting midline crossing by inhibiting Robo1 activity that escapes the commissureless routing effects. The authors characterize this role for Robo2 by examining midline crossing defects in the sensitized background where Netrin-Frazzled signaling is disrupted. The authors demonstrate this effect can occur non-cell autonomously, and provide evidence that midline cells express Robo2 at the correct time of development for it to inhibit Robo1-Slit biology. The authors use domain deletions to show that this effect of Robo2 on promoting crossing is independent of its Slit-binding properties, separating it from the requirements for Robo2 as a midline repellant. The authors also demonstrate binding of Robo2 to Robo1, with some evidence for a trans interaction, and suggest a potential pathway by which Robo2 expressed on midline cells inhibits residual Robo1 on axons to facilitate midline crossing. Among the many features of the paper that are quite compelling is the demonstration that midline crossing by Robo2 does not require its intracellular domain, which likely eliminates a signal-transduction role for Robo2 as a receptor for an unknown positive attractor. This points to an inhibitory role for Robo2. The question then becomes how Robo2 acts. One model would be a cis-inhibition as found for Robo3 in mammalian commissural axons, where it blocks the function of the other Robos. The authors provide data for an alternative model that Robo2 instead acts in midline cells to non-cell autonomously silence Robo1 in a trans interaction. In summary, the data are thorough and the data strongly support new ideas concerning the mechanisms that shape complex neural circuitry during embryonic development. As such, the paper is considered appropriate for the general audience of *eLife*. However, there are a few experimental concerns that should be addressed. Finally, given the general audience targeted by *eLife*, the manuscript needs to be re-written to enable the general readership to better understand access the data and logic presented in this paper.

1) Throughout the manuscript (Figures 2, 3 and 5 in particular), comparisons are made between genotypes expressing different transgenes with conclusions being made regarding the extent of rescue. The authors have gone the extra length to control for differential transgene expression by integrating transgenes at a single expression locus. This is considered excellent. However, concerns were raised by the reviewers that this does not account for the possibility that mutant proteins fail to reach the cell surface. The reviewers acknowledge that it is not possible to make such a determination in the embryo with available reagents. However, controls in heterologous cells to tests that transgenic proteins are processed and traffic correctly would be a nice addition to the manuscript. This experiment would be able to determine gross changes in the trafficking of mutant proteins and would not be expected to measure subtle differences, as this could not be directly compared to the gene expression in vivo.

2) Slit binding should be quantified, if possible. The S2 cells shown in Figure 4 could indicate that Slit binding is reduced, or it could be abolished depending on the imaging methods. The differences in magnitude could be quite important for the conclusions of the manuscript. Along this line, an experiment should be performed to test Slit-Robo binding in heterologous cells transfected with either Robo1 alone or with Robo1 and Robo2.

3) In the subsection headed “Robo2 binds to Robo1 in vivo and the interaction depends on Ig1 and Ig2”, the authors emphasize a trans-signaling interaction. More specifically, the conclusion that Robo1 and Robo2 interact in trans should be tempered (“Although we detected strong interactions between Robo1 and Robo2 in co-transfected cells, we could not detect interactions in cells that were transfected separately and mixed together”). This is one possibility supported by the data, yet there are caveats. For example, no interaction is seen in separately transfected cells unless lysates are prepared where cis and trans can no longer be distinguished. There is some robust biochemistry to demonstrate the Robo2-Robo1 interaction, although the evidence for a strictly trans interaction are the weakest of the data in this section.

4) The manuscript should be significantly revised to make it accessible to the non-expert reader given the intended general audience of *eLife*. Even a geneticist who is not immersed in this area of research has a difficult time navigating the Results section. The figures are beautiful and the data are both solid and interesting, but the experimental logic and conclusions could be conveyed more clearly and simply. The authors are encouraged to use schematics in their figures and, most importantly, clarify the text.

5) Many of the figures lack evidence of statistical comparisons in their bar graphs; compare Figure 1 to Figure 2/3, etc. This should be consistent throughout all figures.

---

## [Author Response]

*1) Throughout the manuscript (*Figures 2, 3 and 5
*in particular), comparisons are made between genotypes expressing different transgenes with conclusions being made regarding the extent of rescue. The authors have gone the extra length to control for differential transgene expression by integrating transgenes at a single expression locus. This is considered excellent. However, concerns were raised by reviewers that this does not account for the possibility that mutant proteins fail to reach the cell surface. The reviewers acknowledge that it is not possible to make such a determination in the embryo with available reagents. However, controls in heterologous cells to tests that transgenic proteins are processed and traffic correctly would be a nice addition to the manuscript. This experiment would be able to determine gross changes in the trafficking of mutant proteins and would not be expected to measure subtle differences, as this could not be directly compared to the gene expression in vivo*.

We thank the reviewers for raising this important point. In the original Figure 4, all three of the deletion variants are robustly expressed and localized to the plasma membrane as detected by their N-terminal HA epitope tags. In addition, since Robo2∆Ig2 is able to bind to bath applied Slit (and mediate Slit-repulsion in vivo), this receptor must be present on the surface. We also note that the ∆Ig1 variant still has significant pro-crossing activity, which is consistent with surface localization. In addition, our western blots show clear bands of the expected sizes for all of the variants that we analyze, suggesting that there are not problems with protein folding.

To further address whether or not our Robo2 deletion variants are trafficked normally in vivo, we have added a supplemental figure (Figure 4—figure supplement 2) demonstrating that all of the receptor variants are localized to CNS axons and are present at comparable levels. Finally, we have performed detergent free staining of S2R+ cells expressing each receptor variant and observe robust surface localization (Figure 4—figure supplement 2).

*2) Slit binding should be quantified, if possible. The S2 cells shown in*
Figure 4
*could indicate that Slit binding is reduced, or it could be abolished depending on the imaging methods. The differences in magnitude could be quite important for the conclusions of the manuscript*.

We have now included pixel intensity quantification of the Slit-binding presented in Figure 4 and have included this data as a new supplemental figure (Figure 4—figure supplement 1). We find that the Robo2∆Ig1 and Robo2∆Ig1-2 do not detectably bind Slit in this assay. The Slit signal detected on these cells is no different from that observed on untransfected cells. In addition, we note a slight reduction in the ability of Robo2∆Ig2 to bind Slit. The biological meaning of this, if any, is unclear, especially given that Robo2∆Ig2 is equivalent to wild-type Robo2 in its ability to mediate repulsion in response to Slit in vivo. This is true when Robo2∆Ig2 is expressed as a UAS transgene and when Robo2∆Ig2 is expressed from a BAC. In order to more rigorously compare any differences in Slit binding due to deletion of Ig2, more quantitative biochemical/biophysical experiments would be needed. We believe that the key observation here is that as predicted from the structural data, deleting Robo2’s Ig1 domain prevents Slit binding.

*Along this line, an experiment should be performed to test Slit-Robo binding in heterologous cells transfected with either Robo1 alone or with Robo1 and Robo2*.

As requested, we have performed this experiment and are including the findings as a figure for the reviewers (included at the end of this document). We do not see any obvious differences in the ability of Slit to bind cells that co-express Robo1 and Robo2 from cells that are individually transfected. However, we do not believe that this is a clearly interpretable experiment. Even if Robo2 were able to prevent Slit binding to Robo1, the fact that Robo2 itself is a strong Slit binding receptor would complicate the interpretation of this experiment. In order to rigorously address this issue, future competition binding experiments using purified proteins and surface plasmon resonance will be needed. We feel that such analysis is beyond the scope of the current manuscript and that given the caveats to interpreting the cell binding in co-transfected cells, we would rather not include this data in the manuscript.

*3) In the subsection headed “Robo2 binds to Robo1 in vivo and the interaction depends on Ig1 and Ig2”, the authors emphasize a trans-signaling interaction. More specifically, the conclusion that Robo1 and Robo2 interact in trans should be tempered (“Although we detected strong interactions between Robo1 and Robo2 in co-transfected cells, we could not detect interactions in cells that were transfected separately and mixed together”). This is one possibility supported by the data, yet there are caveats. For example, no interaction is seen in separately transfected cells unless lysates are prepared where cis and trans can no longer be distinguished. There is some robust biochemistry to demonstrate the Robo2-Robo1 interaction, although the evidence for a strictly trans interaction are the weakest of the data in this section*.

We have modified the text to more clearly emphasize this important caveat.

*4) The manuscript should be significantly revised to make it accessible to the non-expert reader given the intended general audience of* eLife*. Even a geneticist who is not immersed in this area of research has a difficult time navigating the Results section. The figures are beautiful and the data are both solid and interesting, but the experimental logic and conclusions could be conveyed more clearly and simply. The authors are encouraged to use schematics in their figures and, most importantly, clarify the text.*

We have modified the text of the Results section to simplify and clarify the major findings.

*5) Many of the figures lack evidence of statistical comparisons in their bar graphs; compare*
Figure 1
*to Figure 2/3, etc. This should be consistent throughout all figures*.

We have now included statistical comparisons in all of the figures and report *p*-values and statistical methods in the figure legends.